# IBD risk loci are enriched in multigenic regulatory modules encompassing putative causative genes

Yukihide Momozawa, Julia Dmitrieva et al.[#]

GWAS have identified >200 risk loci for Inflammatory Bowel Disease (IBD). The majority of disease associations are known to be driven by regulatory variants. To identify the putative causative genes that are perturbed by these variants, we generate a large transcriptome data set (nine disease-relevant cell types) and identify 23,650 cis-eQTL. We show that these are determined by ~9720 regulatory modules, of which ~3000 operate in multiple tissues and ~970 on multiple genes. We identify regulatory modules that drive the disease association for 63 of the 200 risk loci, and show that these are enriched in multigenic modules. Based on these analyses, we resequence 45 of the corresponding 100 candidate genes in 6600 Crohn disease (CD) cases and 5500 controls, and show with burden tests that they include likely causative genes. Our analyses indicate that ≥10-fold larger sample sizes will be required to demonstrate the causality of individual genes using this approach.

#A full list of authors and their affliations appears at the end of the paper.

Genome Wide Association Studies (GWAS) scan the entire genome for statistical associations between common variants and disease status in large case–control cohorts. GWAS have identified tens to hundreds of risk loci for nearly all studied common complex diseases of human[1]. The study of Inflammatory Bowel Disease (IBD) has been particularly successful, with more than 200 confirmed risk loci reported to date[2,3]. As a result of the linkage disequilibrium (LD) patterns in the human genome (limiting the mapping resolution of association studies), GWAS-identified risk loci typically span ~250 kb, encompassing an average of ~5 genes (numbers ranging from zero ("gene deserts") to more than 50) and hundreds of associated variants. Contrary to widespread misconception, the causative variants and genes remain unknown for the vast majority of GWAS-identified risk loci. Yet, this remains a critical goal in order to reap the full benefits of GWAS in identifying new drug targets and developing effective predictive and diagnostic tools. It is the main objective of post-GWAS studies.

Distinguishing the few causative variants (i.e., the variants that are directly causing the gene perturbation) from the many neutral variants that are only associated with the disease because they are in LD with the former in the studied population, requires the use of sophisticated fine-mapping methods applied to very large, densely genotyped data sets[4], ideally followed-up by functional studies[5]. Using such approaches, 18 causative variants for IBD were recently fine-mapped at single base pair resolution, and 51 additional ones at ≤10 base pair resolution[4].

A minority of causative variants are coding, i.e., they alter the amino-acid sequence of the encoded protein. In such cases, and particularly if multiple such causative coding variants are found in the same gene (i.e., in case of allelic heterogeneity), the corresponding causative gene is unambiguously identified. In the case of IBD, causative genes have been identified for approximately ten risk loci on the basis of such "independently" (i.e., not merely reflecting LD with other variants) associated coding variants, including *NOD2*, *ATG16L1*, *IL23R*, *CARD9*, *FUT2*, and *TYK2*[4,6–9].

For the majority of risk loci, the GWAS signals are not driven by coding variants. They must therefore be driven by common regulatory variants, i.e., variants that perturb the expression levels of one (or more) target genes in one (or more) disease relevant cell types[4]. Merely reflecting the proportionate sequence space that is devoted to the different layers of gene regulation (transcriptional, posttranscriptional, translational, posttranslational), the majority of regulatory variants are likely to perturb components of "gene switches" (promoters, enhancers, insulators), hence affecting transcriptional output. Indeed, fine-mapped noncoding variants are enriched in known transcription-factor binding sites and epigenetic signatures marking gene switch components[4]. Hence, the majority of common causative variants underlying inherited predisposition to common complex diseases must drive *cis*-eQTL (expression quantitative trait loci) affecting the causative gene(s) in one or more disease relevant cell types. The corresponding *cis*-eQTL are expected to operate prior to disease onset, and—driven by common variants—detectable in cohorts of healthy individuals of which most will never develop the disease. The term *cis*-eQTL refers to the fact that the regulatory variants that drive them only affect the expression of genes/alleles residing on the same DNA molecule, typically no more than one megabase away. Causative variants, whether coding or regulatory, may secondarily perturb the expression of genes/alleles located on different DNA molecules, generating *trans*-eQTL. Some of these *trans*-eQTL may participate in the disease process.

*cis*-eQTL effects are known to be very common, affecting more than 50% of genes[10]. Hence, finding that variants associated with a disease are also associated with changes in expression levels of a neighboring gene is not sufficient to incriminate the corresponding genes as causative. Firstly, one has to show that the local association signal for the disease and for the eQTL are driven by the same causative variants. A variety of "colocalisation" methods have been developed to that effect[11–13]. Secondly, regulatory variants may affect elements that control the expression of multiple genes[14], which may not all contribute to the development of the disease, i.e., be causative. Thus, additional evidence is needed to obtain formal proof of gene causality. In humans, the only formal test of gene causality that is applicable is the family of "burden" tests, i.e., the search for a differential burden of disruptive mutations in cases and controls, which is expected only for causative genes[15]. Burden tests rely on the assumption that—in addition to the common, mostly regulatory variants that drive the GWAS signal—the causative gene will be affected by low frequency and rare causative variants, including coding variants. Thus, the burden test makes the assumption that allelic heterogeneity is common, which is supported by the pervasiveness of allelic heterogeneity of Mendelian diseases in humans[16]. Burden tests compare the distribution of rare coding variants between cases and controls[15]. The signal-to-noise ratio of the burden test can be increased by restricting the analysis to coding variants that have a higher probability to disrupt protein function[15]. In the case of IBD, burden tests have been used to prove the causality of *NOD2*, *IL23R*, and *CARD9*[6,8,9]. A distinct and very elegant genetic test of gene causality is the reciprocal hemizygosity test, and the related quantitative complementation assay[17,18]. However, with few exceptions[19,20], it has only been applied in model organisms in which gene knock-outs can be readily generated[21].

In this paper, we describe the generation of a new and large data set for eQTL analysis (350 healthy individuals) in nine cell types that are potentially relevant for IBD. We identify and characterize ~24,000 *cis*-eQTL. By comparing disease and eQTL association patterns (EAP) using a newly developed statistic, we identify 99 strong positional candidate genes in 63 GWAS-identified risk loci. We resequence the 555 exons of 45 of these in 6600 cases and 5500 controls in an attempt to prove their causality by means of burden tests. The outcome of this study is relevant to post-GWAS studies of all common complex disease in humans.

## Results

**Clustering *cis*-eQTL into regulatory modules**. We generated transcriptome data for six circulating immune cell types (CD4+ T lymphocytes, CD8+ T lymphocytes, CD19+ B lymphocytes, CD14+ monocytes, CD15+ granulocytes, platelets) as well as ileal, colonic, and rectal biopsies (IL, TR, RE), collected from 323 healthy Europeans (141 men, 182 women, average age 56 years, visiting the clinic as part of a national screening campaign for colon cancer) using Illumina HT12 arrays (CEDAR data set; Methods). IBD being defined as an inappropriate mucosal immune response to a normal commensal gut flora[22], these nine cell types can all be considered to be potentially disease relevant. Using standard methods based on linear regression and two megabase windows centered on the position of the interrogating probe (Methods), we identified significant *cis*-eQTL (FDR < 0.05) for 8804 of 18,580 tested probes (corresponding to 7216 of 13,615 tested genes) in at least one tissue, amounting to a total of 23,650 *cis*-eQTL effects (Supplementary Data 1). When a gene shows a *cis*-eQTL in more than one tissue, the corresponding "eQTL association patterns" (EAP) (i.e., the distribution of association −log(p) values for all the variants in the region of interest) are expected to be similar if determined by the same regulatory variants, and dissimilar otherwise. Likewise, if several

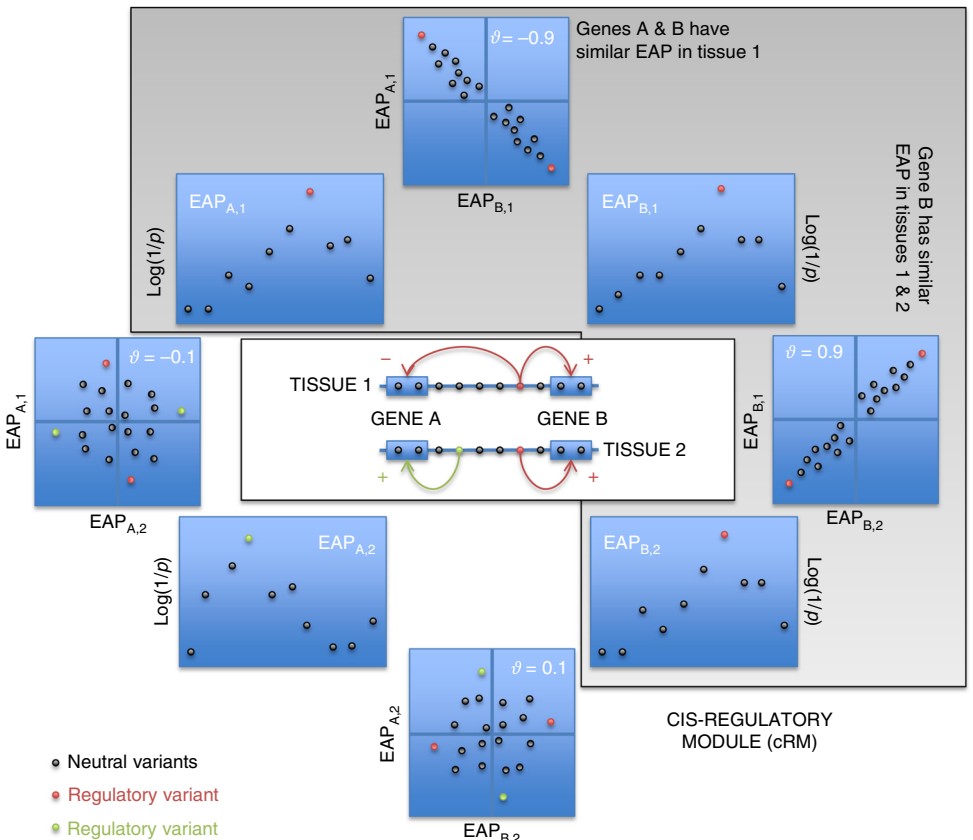

**Fig. 1** *cis*-Regulatory Module (cRM). A *cis*-eQTL affecting gene A in tissue 1 reveals itself by an "eQTL Association Pattern" (EAP$_{A,1}$), i.e., the pattern of –log (*p*) values for variants in the region. Multiple EAP can be observed in a given chromosome region, affecting one or more genes in one or more cell types. EAP that are driven by the same underlying variants are expected to be similar, while EAP driven by distinct variants (for instance, the green and red regulatory variants in the figure) are not. Based on the measure of similarity introduced in this work, $\vartheta$, we cluster the EAP in cRM. For EAP in the same module, $\vartheta$ can be positive or negative, indicating that the variants have the same sign of effect (increasing or decreasing expression) for the corresponding EAP pair

neighboring genes show *cis*-eQTL in the same or distinct tissues, the corresponding EAP are expected to be similar if determined by the same regulatory variants, and dissimilar otherwise (Fig. 1). We devised the $\vartheta$ metric to measure the similarity between association patterns (Methods). $\vartheta$ is a correlation measure for paired $-\log(p)$ values (for the two eQTL that are being compared) that ranges between $-1$ and $+1$. $\vartheta$ shrinks to zero if Pearson's correlation between paired $-\log(p)$ values does not exceed a chosen threshold (i.e., if the EAP are not similar). $\vartheta$ approaches $+1$ when the two EAP are similar and when variants that increase expression in eQTL 1 consistently increase expression in eQTL 2. $\vartheta$ approaches $-1$ when the two EAP are similar and when variants that increase expression in eQTL 1 consistently decrease expression in eQTL 2. $\vartheta$ gives more weight to variants with high $-\log(p)$ for at least one EAP (i.e., it gives more weight to eQTL peaks). Based on the known distribution of $\vartheta$ under $H_0$ (i.e., eQTL determined by distinct variants in the same region) and $H_1$ (i.e., eQTL determined by the same variants), we selected a threshold value $|\vartheta| > 0.60$ to consider that two EAP were determined by the same variant. This corresponds to a false positive rate of 0.05, and a false negative rate of 0.23 (Supplementary Fig. 1). We then grouped EAP in "*cis*-acting regulatory modules" (cRM) using $|\vartheta|$ and a single-link clustering approach (i.e., an EAP needs to have $|\vartheta| > 0.60$ with at least one member of the cluster to be assigned to that cluster). Clusters were visually examined and 29 single edges connecting otherwise unlinked and yet tight clusters manually removed (Supplementary Fig. 2).

Using this approach, we clustered the 23,650 effects in 9720 distinct "*cis*-regulatory modules" (cRM), encompassing *cis*-eQTL with similar EAP (Supplementary Data 2). Sixty-eight percent of cRM were gene- and tissue-specific, 22% were gene-specific but operating across multiple tissues (≤9 tissues, average 3.5), and 10% were multi-genic (≤11 genes, average 2.5) and nearly always multi-tissue (Figs. 2 and 3, Supplementary Fig. 2). In this, cRM are considered gene specific if the EAPs in the cluster concern only one gene, and tissue specific if the EAP in the cluster concern only one of the nine cell types. They are, respectively, multigenic and multi-tissue otherwise. cRM operating across multiple tissues tended to affect multiple genes ($r = 0.47$; $p < 10^{-6}$). In such cRM, the direction of the effects tended to be consistent across tissues and genes ($p < 10^{-6}$). Nevertheless, we observed at least 55 probes with effect of opposite sign in distinct cell types ($\vartheta \leq -0.9$), i.e., the corresponding regulatory variants increases transcript levels in one cell type while decreasing them in another (Fig. 4 and Supplementary Data 3). Individual tissues allowed for the detection of 7–33% of all cRM, and contributed 3–14% unique cRM (Supplementary Fig. 3). Sixty-nine percent of cRM were only detected in one cell type. The rate of cRM sharing between cell types reflects known ontogenic relations. Considering cRM shared by only two cell types (i.e., what jointly differentiates these two cell types from all other), revealed the close proximity of the CD4–CD8, CD14–CD15, ileum–colon, and colon–rectum pairs. Adding information of cRM shared by up to six cell types grouped lymphoid (CD4, CD8, CD19), myeloid (CD14, CD15 but not platelets), and intestinal (ileum, colon and rectum) cells.

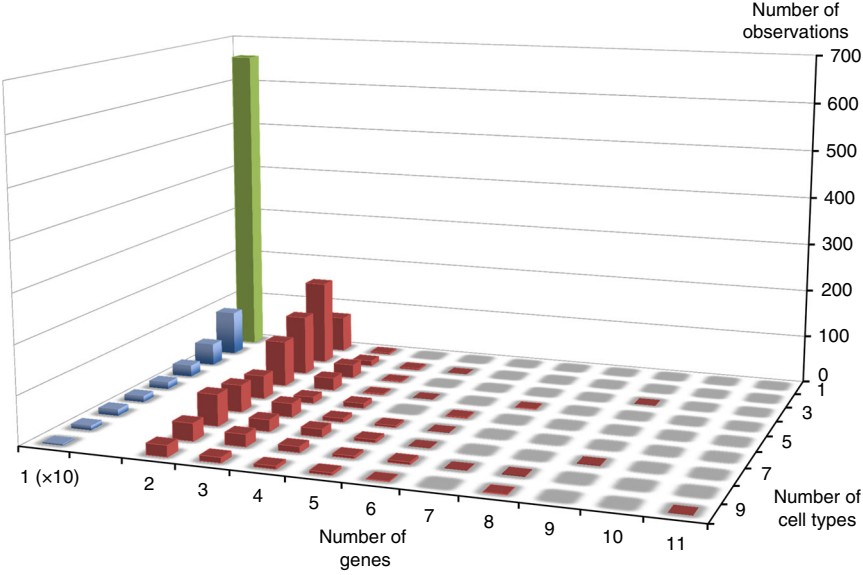

**Fig. 2** Single-gene/tissue versus multi-gene/tissue cRM. Using $|\vartheta| > 0.6$, the 23,950 *cis*-eQTL (FDR $\leq 0.05$) detected in the nine analyzed cell types were clustered in 9720 *cis*-Regulatory Modules (cRM). 68% of these were single-gene, single-tissue cRM (green), 22% were single-gene, multi-tissue cRM (blue), and 10% were multi-gene, mostly multi-tissue cRM (red). The number of observations for single-gene cRM were divided by 10 in the graph for clarity. Thus, there are more cases of single-gene, multi-tissue cRM (blue; 2155) than multi-gene cRM (red; 967)

Adding cRM with up to nine cell types revealed a link between ileum and blood cells, possibly reflecting the presence of blood cells in the ileal biopsies (Fig. 5).

**cRM matching IBD association signals are often multigenic.** If regulatory variants affect disease risk by perturbing gene expression, the corresponding "disease association patterns" (DAP) and EAP are expected to be similar, even if obtained in distinct cohorts (yet with same ethnicity) (Fig. 6). We confronted DAP and EAP using the $\vartheta$ statistic and threshold ($|\vartheta| > 0.60$) described above for 200 GWAS-identified IBD risk loci. DAP for Crohn's disease and ulcerative colitis were obtained from the International IBD Genetics Consortium (IIBDGC)[2,3], EAP from the CEDAR data set.

The probability that two unrelated association signals in a chromosome region of interest are similar (i.e., have high $|\vartheta|$ value) is affected by the degree of LD in the region. If the LD is high it is more likely that two association signals are similar by chance. To account for this, we generated EAP- and locus-specific distributions of $|\vartheta|$ by simulating eQTL explaining the same variance as the studied eQTL, yet driven by 100 variants that were randomly selected in the risk locus (matched for MAF), and computing $|\vartheta|$ with the DAP for all of these. The resulting empirical distribution of $|\vartheta|$ was used to compute the probability to obtain a value of $|\vartheta|$ as high or higher than the observed one, by chance alone (Methods).

Strong correlations between DAP and EAP ($|\vartheta| > 0.6$, associated with low empirical *p* values) were observed for at least 63 IBD risk loci, involving 99 genes (range per locus: 1–6) (Table 1, Fig. 7, Supplementary Data 4). Increased disease risk was associated equally frequently with increased as with decreased expression ($p_{CD} = 0.48$; $p_{UC} = 0.88$). An open-access website has been prepared to visualize correlated DAP–EAP within their genomic context (http://cedar-web.giga.ulg.ac.be). Genes with highest $|\vartheta|$ values ($\geq 0.9$) include known IBD causative genes (for instance, *ATG16L1*, *CARD9*, and *FUT2*), known immune regulators (for instance, *IL18R1*, *IL6ST*, and *THEMIS*), as well as genes with as of yet poorly defined function in the context of IBD (for instance, *APEH*, *ANKRD55*, *CISD1*, *CPEB4*, *DOCK7*,

*ERAP2*, *GNA12*, *GPX1*, *GSDMB*, *ORMDL3*, *SKAP2*, *UBE2L3*, and *ZMIZ1*) (Supplementary Note 1).

The eQTL link with IBD has not been reported before for at least 47 of the 99 reported genes (Table 1). eQTL links with IBD have been previously reported for 111 additional genes, not mentioned in Table 1. Our data support these links for 19 of them, however, with $|\vartheta| \leq 0.6$ (Supplementary Data 5). We applied SMR[13] as alternative colocalisation method to our data. Using a Bonferroni-corrected threshold of $\leq 2.5 \times 10^{-5}$ for $p_{SMR}$ and $\geq 0.05$ for $p_{HEIDI}$, SMR detected 35 of the 99 genes selected with $\vartheta$ (Supplementary Data 4). Using the same thresholds, SMR detected nine genes that were not selected by $\vartheta$. Of these, three (*ADAM15*, *AHSA2*, *UBA7*) had previously been reported by others, while six (*FAM189B*, *QRICH1*, *RBM6*, *TAP2*, *ADO*, *LGALS9*) were not. Of these six, three (*RBM6*, *TAP2*, *ADO*) were characterized by $0.45 < |\vartheta| < 0.6$ (Supplementary Data 5).

Using an early version of the CEDAR data set, significant (albeit modest) enrichment of overlapping disease and eQTL signals was reported for CD4, ileum, colon and rectum, focusing on 76 of 97 studied IBD risk loci (MAF of disease variant >0.05)[4]. By pre-correcting fluorescence intensities with 23 to 53 (depending on cell type) principal components to account for unidentified confounders (Methods), we increased the number of significant eQTL from 480 to 880 in the corresponding 97 regions (11,964 to 23,650 for the whole genome). We repeated the enrichment analysis focusing on 63 of the same 97 IBD loci (CD risk loci; MAF of disease variant >0.05), using three colocalisation methods including $\vartheta$ (Methods). We observed a systematic excess overlap in all analyzed cell types (2.5-fold on average). The enrichment was very significant with the three methods in CD4 and CD8 (Supplementary Table 1).

The 400 analyzed DAP (200 CD and 200 UC) were found to match 76 cRM (in 63 risk loci) with $|\vartheta| > 0.6$ (Table 1), of which 25 are multigenic. Knowing that multigenic cRM represent 10% of all cRM (967/9720), 25/76 (i.e., 33%) corresponds to a highly significant three-fold enrichment ($p < 10^{-9}$). To ensure that this apparent enrichment was not due to the fact that multigenic cRM have more chance to match DAP (as by definition multiple EAP are tested for multigenic cRM), we repeated the enrichment

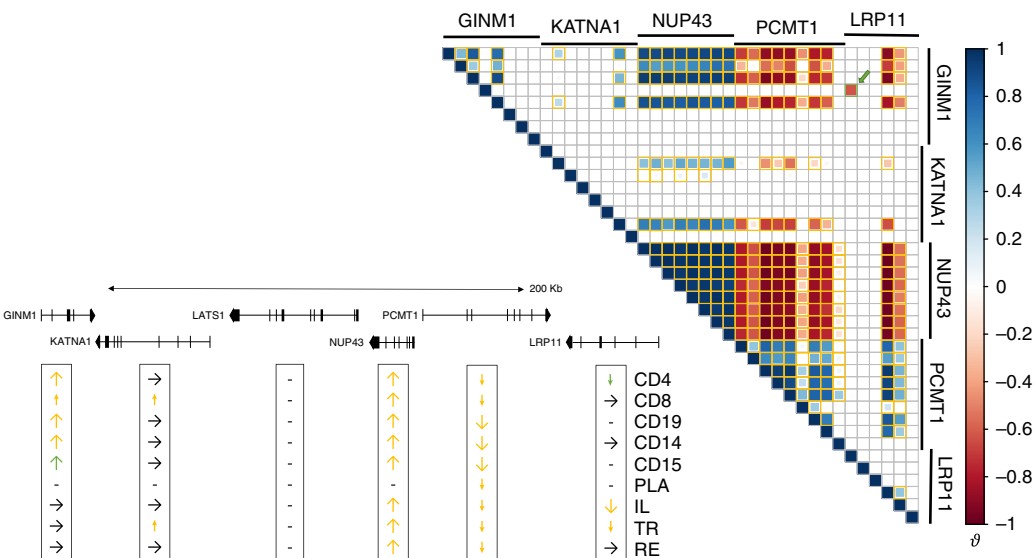

**Fig. 3** Example of a multi-gene, multi-tissue cRM. Gene-tissue combinations for which no expression could be detected are marked by "-", with detectable expression but without evidence for *cis*-eQTL as " → ", with detectable expression and evidence for a *cis*-eQTL as "↑" or "↓" (large arrows: FDR < 0.05; small arrows: FDR ≥ 0.05 but high |ϑ| values). eQTL labeled by the yellow arrows constitute the multi-genic and multi-tissular cRM no. 57. The corresponding regulatory variant(s) increase expression of the *GINM1*, *NUP43* and probably *KATNA1* genes (left side of the cRM), while decreasing expression of the *PCMT1* and *LRP11* genes (right side of the cRM). The expression of *GINM1* in CD15 and *LRP11* in CD4 appears to be regulated in opposite directions by a distinct cRM (no. 3694, green). The *LATS1* gene, in the same region, is not affected by the same regulatory variants in the studied tissues. Inset 1: ϑ values for all EAP pairs. EAP pairs with |ϑ| > 0.6 are bordered in yellow when corresponding to cRM no. 57, in green when corresponding to cRM no. 3694 (+green arrow)

analysis by randomly sampling only one representative EAP per cRM in the 200 IBD risk loci. The frequency of multigenic cRM amongst DAP-matching cRM averaged 0.22, and was never ≤0.10 ($p \leq 10^{-5}$) (Supplementary Fig. 4). In loci with high LD, EAP driven by distinct regulatory variants (yet in high LD) may erroneously be merged in the same cRM. To ensure that the observed enrichment in multigenic cRM was not due to higher levels of LD, we compared the LD-based recombination rate of the 63 cRM-matching IBD risk loci with that of the rest of the genome (https://github.com/joepickrell/1000-genomes-genetic-maps). The genome-average recombination rate was 1.23 centimorgan per megabase (cM/Mb), while that of the 63 IBD risk loci was 1.34 cM/Mb, i.e., less LD in the 63 cRM-matching IBD risk loci than in the rest of the genome. We further compared the average recombination rate in the 63 cRM-matching IBD regions with that of sets of 63 loci centered on randomly drawn cRM (from the list of 9720), matched for size and chromosome number (as cM/Mb is affected by chromosome size). The average recombination rate around all cRM was 1.43 cM/Mb, and this didn't differ significantly from the 63 cRM-matching IBD regions ($p = 0.46$) (Supplementary Fig. 5). Therefore, the observed enrichment cannot be explained by a higher LD in the 63 studied IBD risk loci. Taken together, EAP that are strongly correlated with DAP ($|ϑ| \geq 0.60$), map to regulatory modules that are 2- to 3-fold enriched in multigenic cRM when compared to the genome average and include four of the top 10 (of 9720) cRM ranked by number of affected genes.

**DAP-matching cRM are enriched in causative genes for IBD.** For truly causative genes, the burden of rare disruptive variants is expected to differ between cases and controls[23]. We therefore performed targeted sequencing for the 555 coding exons (~88 kb) of 38 genes selected amongst those with strongest DAP–EAP

correlations, plus seven genes with suggestive DAP–EAP evidence backed by literature (Table 1), in 6597 European CD cases and 5502 matched controls (ref.[24] and Methods). Eighteen of these were part of single-gene cRM and the only gene highlighted in the corresponding locus. The remaining 27 corresponded to multi-gene cRM mapping to 15 risk loci. We added the well-established *NOD2* and *IL23R* causative IBD genes as positive controls. We identified a total of 174 loss-of-function (LoF) variants, 2567 missense variants (of which 991 predicted by SIFT[25] to be damaging and Polyphen-2[26] to be either possibly or probably damaging), and 1434 synonymous variants (Fig. 8 and Supplementary Data 6). 1781 of these were also reported in the Genome Aggregation Database[27] with nearly identical allelic frequencies (Supplementary Fig. 6). We designed a gene-based burden test to simultaneously evaluate hypothesis (i): all disruptive variants enriched in cases (when $ϑ < 0$; risk variants) or all disruptive variants enriched in controls (when $ϑ > 0$; protective variants), and hypothesis (ii): some disruptive variants enriched in cases and others in controls. Hypothesis (i) was tested with CAST[28], and hypothesis (ii) with SKAT[29] (Methods). We restricted the analysis to 1141 LoF and damaging missense variants with minor allele frequency (MAF) ≤0.005 to ensure that any new association signal would be independent of the signals from common and low frequency variants having led to the initial identification and fine-mapping of the corresponding loci[4]. For *NOD2* ($p = 6.9 \times 10^{-7}$) and *IL23R* ($p = 1.8 \times 10^{-4}$), LoF and damaging variants were significantly enriched in respectively cases and controls as expected. When considering the 45 newly tested genes as a whole, we observed a significant ($p = 6.9 \times 10^{-4}$) shift towards lower p values when compared to expectation, while synonymous variants behaved as expected ($p = 0.66$) (Fig. 9 and Supplementary Data 7). This strongly suggests that the sequenced list includes causative genes. *CARD9*, *TYK2*, and *FUT2* have recently been shown to be causative genes based on disease-associated low-frequency coding variants (MAF > 0.005)[4]. The

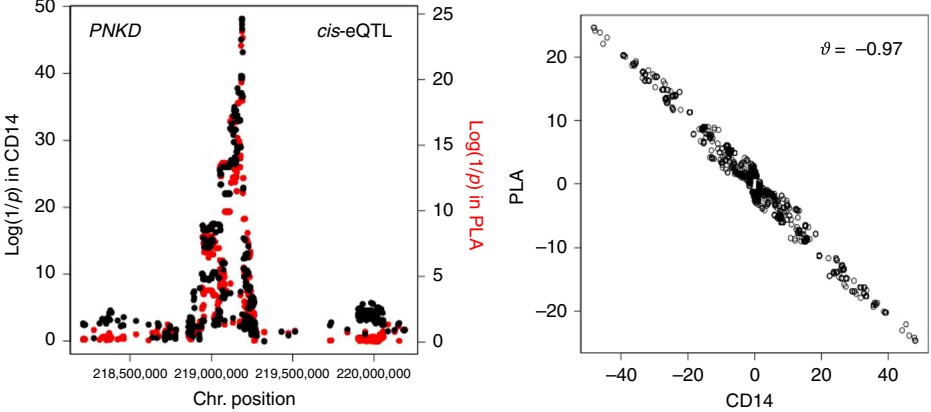

**Fig. 4** Variant(s) with opposite effects on expression in two cell types. Example of a gene (*PNKD*) affected by a *cis*-eQTL in at least two cell types (CD14 and platelets) that are characterized by EAP with $\vartheta = -0.97$, indicating that the gene's expression level is affected by the same regulatory variant in these two cell types, yet with opposite effects, i.e., the variant that is increasing expression in platelets is decreasing expression in CD14

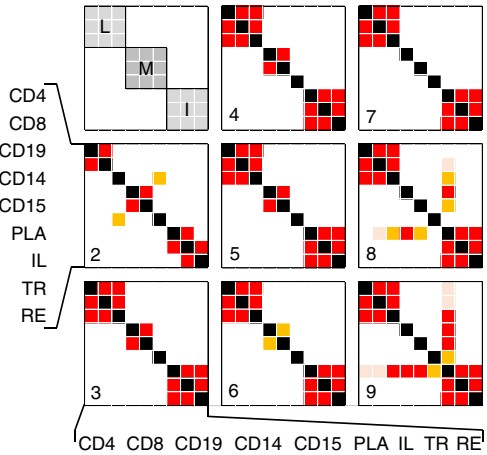

**Fig. 5** Significance of the excess sharing of cRM between cell types. (red: $p < 0.0002$ (Bonferroni corrected 0.0144), orange: $p < 0.001$ (0.072), rose: $p < 0.01$ (0.51)). The numbers in the lower-left corner of the squares indicate which cRM were used for the analysis: (2) cRM affecting no more than two cell types, (3) cRM affecting no more than three cell types, etc. The upper-left square indicates the position of the lymphoid cell types (L) (CD4, CD8, CD19), the myeloid cell types (M)(CD14, CD15, PLA), and the intestinal cell types (I)(IL, TR, RE). For each pair of cell types $i$ and $j$, we computed two $p$ values, one using $i$ as reference, the other using $j$ as reference (Methods). Pairs of $p$ values were always consistent

shift towards lower $p$ values remained significant without these ($p = 1.7 \times 10^{-3}$), pointing towards novel causative genes amongst the 42 remaining candidate genes.

**Proving gene causality requires larger case–control cohorts**. Despite the significant shift towards lower $p$ values when considering the 45 genes jointly, none of these were individually significant when accounting for multiple testing ($p \leq \frac{0.05}{2*45} \approx 0.0006$) (Supplementary Data 7). Near identical results were obtained when classifying variants using the Combined Annotation Dependent Depletion (CADD) tool[30] instead of SIFT/PolyPhen-2 (Supplementary Data 7). We explored three approaches to increase the power of the burden test. The first built on the observation that cRM matching DAP are enriched in multigenic modules. This suggests that part of IBD risk loci harbor multiple co-regulated and hence functionally related

genes, of which several (rather than one, as generally assumed) may be causally involved in disease predisposition. To test this hypothesis, we designed a module- rather than gene-based burden test (Methods). However, none of the 30 tested modules reached the experiment-wide significance threshold ($p \leq \frac{0.05}{2*30} \approx 0.0008$). Moreover, the shift towards lower $p$ values for the 30 modules was not more significant ($p = 2.3 \times 10^{-3}$) than for the gene-based test (Supplementary Fig. 7a and Supplementary Table 7). The second and third approaches derive from the common assumption that the heritability of disease predisposition may be larger in familial and early-onset cases[31]. We devised orthogonal tests for age-of-onset and familiality and combined them with our burden tests (Methods). Neither approach would improve the results (Supplementary Fig. 7b, c and Supplementary Data 7).

Assuming that *TYK2* and *CARD9* are truly causative and their effect sizes in our data unbiased, we estimated that a case–control cohort ranging from ~50,000 (*TYK2*) to ~200,000 (*CARD9*) individuals would have been needed to achieve experiment-wide significance (testing 45 candidate genes), and from ~78,000 (*TYK2*) to >500,000 (*CARD9*) individuals to achieve genome-wide significance (testing 20,000 genes) in the gene-based burden test (Supplementary Fig. 8).

## Discussion

We herein describe a novel dataset comprising array-based transcriptome data for six circulating immune cell types and intestinal biopsies at three locations collected on ~300 healthy European individuals. We use this "Correlated Expression and Disease Association Research" data set (CEDAR) to identify 23,650 significant *cis*-eQTL, which fall into 9720 regulatory modules of which at least ~889 affect more than one gene in more than one tissue. We provide strong evidence that 63 of 200 known IBD GWAS signals reflect the activity of common regulatory variants that preferentially drive multigenic modules. We perform an exon-based burden test for 45 positional candidate CD genes mapping to 33 modules, in 5500 CD cases and 6500 controls. By demonstrating a significant ($p = 6.9 \times 10^{-4}$) upwards shift of log(1/$p$) values for damaging when compared to synonymous variants, we show that the sequenced genes include new causative CD genes.

Individually, none of the sequenced genes (other than the positive *NOD2* and *IL23R* controls) exceed the experiment-wide significance threshold, precluding us from definitively pinpointing any novel causative genes. However, we note *IL18R1* amongst

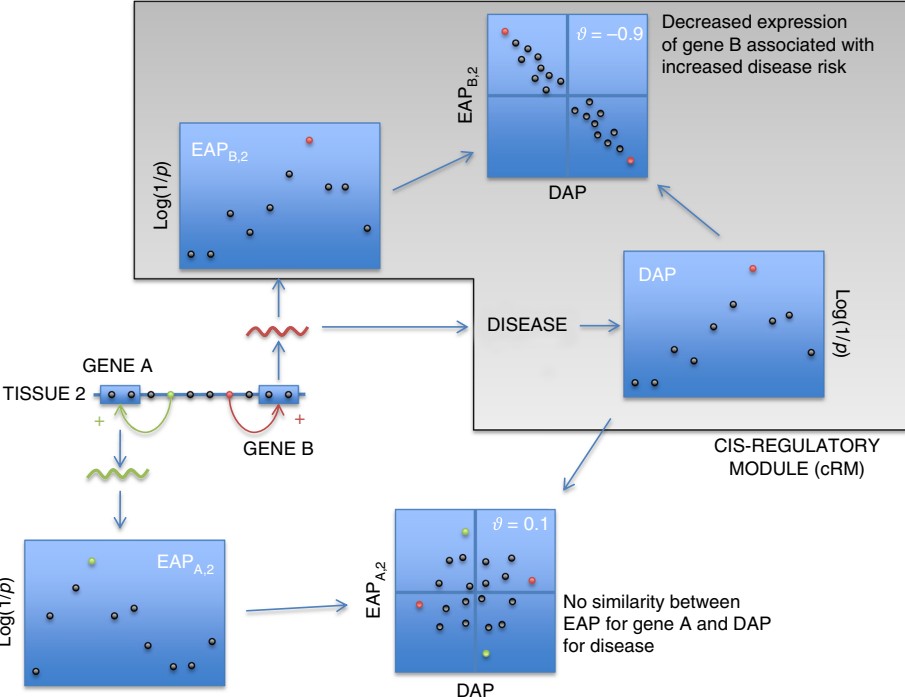

**Fig. 6** DAP-matching cRM. If a regulatory variant (red) affects disease risk by altering the expression levels of gene B in tissue 2, the $EAP_{B,2}$ is expected to be similar (high $|\vartheta|$) to the "disease association pattern" (DAP), both assigned therefore to the same cRM. $\vartheta$ is positive if increased gene expression is associated with increased disease risk, negative otherwise. A *cis*-eQTL that is driven by a regulatory variant (green) that does not directly affect disease risk, will be characterized by an EAP (say gene A, tissue 2, $EAP_{A,2}$) that is not similar to the DAP (low $|\vartheta|$)

the top-ranking genes (see also Supplementary Note 1). *IL18R1* is the only gene in an otherwise relatively gene-poor region (also encompassing *IL1R1* and *IL18RAP*) characterized by robust *cis*-eQTL in CD4 and CD8 that are strongly correlated with the DAP for CD and UC ($0.68 \leq |\vartheta| \leq 0.93$). Reduced transcript levels of *IL18R1* in these cell types is associated with increased risk for IBD. Accordingly, rare (MAF $\leq 0.005$) damaging variants were cumulatively enriched in CD cases (CAST $p = 0.05$). The cumulative allelic frequency of rare damaging variants was found to be higher in familial CD cases (0.0027), when compared to non-familial CD cases (0.0016; $p = 0.09$) and controls (0.0010; $p = 0.03$). When ignoring carriers of deleterious *NOD2* mutations, average age-of-onset was reduced by ~3 years (25.3 vs 28.2 years) for carriers of rare damaging *IL18R1* variants but this difference was not significant ($p = 0.18$).

While the identification of matching cRM for 63/200 DAP points towards a number of strong candidate causative genes, it leaves most risk loci without matching eQTL despite the analysis of nine disease-relevant cell types. This finding is in agreement with previous reports[4,32]. It suggests that *cis*-eQTL underlying disease predisposition operate in cell types, cell states (for instance, resting vs activated) or developmental stages that were not explored in this and other studies. It calls for the enlargement and extension of eQTL studies to more diverse and granular cellular panels[10,33], possibly by including single-cell sequencing or spatial transcriptomic approaches. By performing eQTL studies in a cohort of healthy individuals, we have made the reasonable assumption that the common regulatory variants that are driving the majority of GWAS signals are acting before disease onset, including in individuals that will never develop the disease. An added advantage of studying a healthy cohort, is that the corresponding dataset is "generic", usable for the study of perturbation of gene regulation for any common complex disease. However, it is conceivable that some eQTL underlying increased disease risk only manifest themselves once

the disease process is initiated, for instance as a result of a modified inflammatory status. Thus, it may be useful to perform eQTL studies with samples collected from affected individuals to see in how far the eQTL landscape is affected by disease status.

One of the most striking results of this work is the observation that cRM that match DAP are $\geq 2$-fold enriched in multi-genic modules. We cannot fully exclude that this is due to ascertainment bias. As multi-genic modules tend to also be multi-tissue, multi-genic cRM matching a DAP in a non-explored disease-relevant cell type have a higher probability to be detected in the explored cell types than the equivalent monogenic (and hence more likely cell type specific) cRM. The alternative explanation is that cRM matching DAP are truly enriched in multi-genic cRM. It is tempting to surmise that loci harboring clusters of co-regulated, functionally related causative genes have a higher probability to be detected in GWAS, reflecting a relatively larger target space for causative mutations. We herein tested this hypothesis by applying a module rather than gene-based test. Although this did not appear to increase the power of the burden test in this work, it remains a valuable approach to explore in further studies. Supplementary Data 2 provides a list of >900 multigenic modules detected in this work that could be used in this context.

Although we re-sequenced the ORF of 45 carefully selected candidate genes in a total of 5500 CD cases and 6600 controls, none of the tested genes exceeded the experiment-wide threshold of significance. This is despite the fact that we used a one-sided, eQTL-informed test to potentially increase power. Established IBD causative genes used as positive control, *NOD2* and *IL23R*, were positive indicating that the experiment was properly conducted. We were not able to improve the signal strength by considering information about regulatory modules, familiality or age-of-onset. We estimated that $\geq 10$-fold larger sample sizes will be needed to achieve adequate power if using the same approach.

**Table 1 IBD risk loci for which at least one *cis*-eQTL association pattern (EAP) was found to match the disease association pattern (DAP)**

| Loc | Chr | Beg | End | cRM | Nr | Genes with correlated DAP–EAP | Implicated cell types | Best θ CD | Best θ UC | Best p CD | Best p UC | Ref |
|---|---|---|---|---|---|---|---|---|---|---|---|---|
| HD1 | 1 | 2.4 | 2.8 | 271 | 2 | *TNFRSF14* | CD4 CD8 IL TR | −0.74 | −0.79 | 0.02 | 0.03 | 4,48 |
| HD2 | 1 | 7.7 | 8.3 | 2900 | 1 | PARK7 | CD15 TR RE | −0.8 | −0.82 | 0.01 | 0.06 | 48 |
| N_1_62 | 1 | 62.5 | 63.5 | 109 | 3 | **DOCK7** USP1 ATG4C | CD4 CD8 **CD19** CD14 CD15 | −0.9 | 0 | 0.01 | 1.00 | 3 |
| N_1_100 | 1 | 101.0 | 102.0 | 6008 | 1 | SLC30A7 | TR | 0 | −0.71 | 1.00 | 0.06 | |
| J_1_119 | 1 | 120.2 | 120.7 | 9459 | 1 | NOTCH2 | CD19 | 0.68 | 0 | 0.13 | 1.00 | |
| HD14 | 1 | 155.0 | 156.1 | 5 | 8 | GBA | CD4 | −0.65 | 0 | 0.01 | 1.00 | |
| | | | | 238 | 3 | *THBS3* GBA MUC1 | CD14 CD15 TR | 0 | 0.81 | 1.00 | 0.02 | |
| | | | | 4513 | 1 | *THBS3* | CD4 | 0 | 0.66 | 1.00 | 0.02 | |
| HD21 | 1 | 197.3 | 198.0 | 6071 | 1 | DENND1B | CD4 | 0.7 | 0.78 | 0.03 | 0.02 | |
| HD30 | 2 | 62.4 | 62.7 | 3716 | 1 | B3GNT2 | CD8 | −0.63 | 0 | 0.01 | 1.00 | |
| HD35 | 2 | 102.8 | 103.3 | 1132 | 1 | **IL18R1** | **CD4 CD8** | −0.93 | −0.87 | 0.01 | 0.03 | 4 |
| | | | | 8912 | 1 | (IL18RAP) | CD8 | −0.42 | 0 | 0.11 | 0.38 | 4 |
| J_2_197 | 2 | 198.2 | 199.1 | 325 | 2 | MARS2 *PLCL1* | CD4 CD14 | −0.72 | 0 | 0.06 | 1.00 | 2,48 |
| J_2_218 | 2 | 218.9 | 219.4 | 216 | 3 | PNKD GPBAR1 | CD14 TR RE | 0.72 | 0.72 | 0.01 | 0.06 | 2,48 |
| HD43 | 2 | 234.1 | 234.6 | 1177 | 1 | ***ATG16L1*** | CD4 CD8 IL TR RE | 0.94 | 0 | 0.05 | 1.00 | 2,49 |
| N_3_45 | 3 | 46.0 | 47.0 | 2930 | 1 | CCR2 | CD19 | 0.77 | 0 | 0.02 | 1.00 | |
| | | | | 1203 | 1 | CCR2 | CD4 | −0.62 | 0 | 0.07 | 1.00 | |
| | | | | 7768 | 1 | CCR9 | CD19 | 0 | −0.67 | 1.00 | 0.06 | |
| | | | | 6798 | 1 | KLHL18 | CD14 | 0 | −0.68 | 1.00 | 0.03 | |
| HD50 | 3 | 48.4 | 51.4 | 8 | 7 | USP4 | CD19 | 0.64 | 0.63 | 0.06 | 0.07 | 2 |
| | | | | 217 | 3 | ***GPX1 APEH*** IP6K1 | **CD19** CD14 **TR RE** | 0.91 | 0.97 | 0.01 | 0.01 | 2,49 |
| | | | | 122 | 3 | *FAM212A* | CD19 | 0 | 0.61 | 1.00 | 0.05 | |
| J_3_52 | 3 | 52.8 | 53.3 | 3190 | 1 | SFMBT1 | TR RE | 0 | −0.88 | 1.00 | 0.01 | 50 |
| J_4_73 | 4 | 74.6 | 75.1 | 1271 | 1 | CXCL5 | CD4 CD8 CD19 CD14 PLA | 0 | −0.84 | 1.00 | 0.01 | 2 |
| HD60 | 5 | 40.0 | 40.7 | | | (PTGER4) | CD15 | 0 | 0 | 0.28 | 0.15 | 51 |
| HD61 | 5 | 55.4 | 55.5 | 360 | 2 | ***ANKRD55* IL6ST** | **CD4 CD8** | 0.9 | 0 | 0.02 | 1.00 | 4 |
| HD62 | 5 | 72.4 | 72.6 | 6625 | 1 | FOXD1 | IL | −0.74 | 0 | 0.03 | 1.00 | 4 |
| HD63 | 5 | 95.9 | 96.5 | 365 | 2 | ***ERAP2* LNPEP** | **CD4 CD8 CD19 CD14 CD15** PLA **IL TR RE** | 0.94 | 0.71 | 0.01 | 0.02 | 2,4,50 |
| HD65 | 5 | 130.4 | 132.0 | 55 | 4 | (SLC22A4) (SLC22A5) | CD4 CD15 | −0.55 | 0 | 0.06 | 0.07 | 4,52 |
| HD66 | 5 | 141.4 | 141.7 | 2389 | 1 | *NDFIP1* | CD8 PLA | 0.87 | 0.88 | 0.04 | 0.01 | 2 |
| HD67 | 5 | 149.0 | 151.0 | – | – | (IRGM) | – | – | – | – | – | 53 |
| HD71 | 5 | 173.2 | 173.6 | 1349 | 1 | CPEB4 | **CD4** CD8 CD19 **CD14** CD15 PLA TR | −0.92 | 0 | 0.01 | 1.00 | 2,4 |
| J_66_32 | 6 | 32.3 | 32.9 | 7853 | 1 | HLA-DQA2 | IL | 0 | −0.62 | 1.00 | 0.02 | |
| HD76 | 6 | 90.8 | 91.1 | 1404 | 1 | BACH2 | CD4 | 0.67 | 0 | 0.14 | 1.00 | |
| HD78 | 6 | 111.3 | 112.0 | 9603 | 1 | SLC16A10 | IL | 0 | −0.71 | 1.00 | 0.11 | |
| HD80 | 6 | 127.9 | 128.4 | 707 | 2 | **THEMIS** PTPRK | **CD8** | −0.92 | 0 | 0.01 | 1.00 | |
| HD83 | 6 | 167.3 | 167.6 | 1425 | 1 | *RNASET2* | CD4 CD8 CD15 PLA | −0.87 | 0 | 0.02 | 1.00 | 4 |
| J_7_1 | 7 | 2.5 | 3.0 | 2729 | 1 | **GNA12** | **CD19** CD14 TR | 0 | −0.94 | 1.00 | 0.02 | 2 |
| HD84 | 7 | 26.6 | 27.3 | 1441 | 1 | **SKAP2** | **CD4 CD8 CD19** | 0.97 | 0 | 0.01 | 1.00 | 4 |
| HD85 | 7 | 28.1 | 28.3 | 6438 | 1 | *JAZF1* | CD4 | 0.78 | 0 | 0.01 | 1.00 | 2 |
| HD92 | 7 | 128.5 | 128.8 | 401 | 2 | IRF5 *TNPO3* | CD15 IL | 0 | −0.64 | 1.00 | 0.02 | 2,48 |
| | | | | 7046 | 1 | TSPAN33 | CD19 | −0.64 | 0 | 0.01 | 1.00 | |
| N_8_26 | 8 | 26.7 | 27.7 | 5869 | 1 | PTK2B | CD14 | −0.69 | 0 | 0.01 | 1.00 | |
| | | | | 5841 | 1 | TRIM35 | CD4 | 0 | 0.66 | 1.00 | 0.01 | |
| HD106 | 9 | 139.1 | 139.5 | 64 | 4 | ***CARD9*** INPP5E SEC16A SDCCAG3 | CD4 CD8 CD19 CD14 **CD15** IL TR RE | 0.95 | 0.86 | 0.01 | 0.02 | 2,4,50 |
| HD109 | 10 | 30.6 | 30.9 | 1603 | 1 | MTPAP | TR | −0.62 | 0 | 0.11 | 1.00 | |
| HD112 | 10 | 59.8 | 60.2 | 1609 | 1 | ***CISD1*** | **CD4 CD8 CD19 CD14 CD15 TR RE** | 0.94 | 0.83 | 0.04 | 0?01 | 2,4,48 |
| J_10_74 | 10 | 75.4 | 75.9 | 436 | 2 | *VCL* | CD4 CD8 CD19 CD14 RE | 0 | −0.79 | 1.00 | 0.04 | |
| | | | | 4279 | 1 | CAM2KG | CD4 | −0.67 | 0 | 0.04 | 1.00 | |
| HD114 | 10 | 81.0 | 81.2 | 5476 | 1 | **ZMIZ1** | **CD8** | −0.91 | −0.86 | 0.03 | 0.01 | |
| J_10_80 | 10 | 82.0 | 82.5 | 712 | 2 | *TSPAN14* | TR | −0.71 | 0 | 0.01 | 1.00 | |
| | | | | 2216 | 1 | TSPAN14 | CD4 CD14 | 0.76 | 0 | 0.01 | 1.00 | 2 |
| HD116 | 10 | 101.2 | 101.4 | 5439 | 1 | *SLC25A28* | CD14 | −0.61 | 0 | 0.22 | 1.00 | |
| J_11_57 | 11 | 58.1 | 58.6 | 7164 | 1 | ZFP91 | PLA | −0.64 | −0.75 | 0.02 | 0.07 | |
| J_11_59 | 11 | 61.3 | 61.8 | 1670 | 1 | TMEM258 | CD4 CD8 CD19 | 0.83 | 0 | 0.04 | 1.00 | |
| J_11_65 | 11 | 65.4 | 65.9 | 451 | 2 | CTSW FIBP | CD4 CD8 | −0.73 | 0 | 0.01 | 1.00 | 2 |
| HD122 | 11 | 114.2 | 114.6 | 268 | 3 | REXO2 NXPE1 NXPE4 | TR RE | 0 | −0.89 | 1.00 | 0.02 | 4,50 |
| HD123 | 11 | 118.3 | 118.8 | 8200 | 1 | TREH | IL | 0 | 0.7 | 1.00 | 0.05 | |
| HD142 | 14 | 88.2 | 88.7 | 8940 | 1 | GPR65 | CD14 | 0.8 | 0.79 | 0.01 | 0.01 | |
| | | | | 6353 | 1 | (GALC) | CD14 | −0.52 | −0.23 | 0.06 | 0.06 | 4 |
| J_15_40 | 15 | 41.3 | 41.8 | 9109 | 1 | CHP1 | IL | 0.62 | 0 | 0.01 | 1.00 | |

**Table 1** (continued)

| Loc | Chr | Beg | End | cRM | Nr | Genes with correlated DAP–EAP | Implicated cell types | Best $\theta$ | | Best $p$ | | Ref |
|---|---|---|---|---|---|---|---|---|---|---|---|---|
| | | | | | | | | CD | UC | CD | UC | |
| J_16_22 | 16 | 23.6 | 24.1 | 2672 | 1 | PRKCB | CD14 | 0 | 0.64 | 1.00 | 0.05 | 2 |
| HD150 | 16 | 28.2 | 29.1 | 6 | 8 | *TUFM* SBK1 APOBR *SGF29 CLN3 SPNS1* | CD4 CD8 CD19 CD14 CD15 IL TR RE | 0.81 | 0.86 | 0.05 | 0.03 | 4 |
| HD151 | 16 | 30.4 | 31.4 | 2673 | 1 | *RNF40* | CD15 | −0.63 | 0 | 0.02 | 1.00 | |
| | | | | 1886 | 1 | *ITGAL* | CD4 CD8 CD19 | 0 | 0.74 | 1.00 | 0.01 | 54 |
| HD153 | 16 | 68.4 | 68.9 | 1894 | 1 | *ZFP90* | CD4 CD8 CD19 CD14 TR | 0 | 0.83 | 1.00 | 0.07 | 2,48 |
| HD156 | 16 | 85.9 | 86.1 | 3328 | 1 | *IRF8* | TR RE | 0 | 0.72 | 1.00 | 0.01 | |
| HD159 | 17 | 37.3 | 38.3 | 37 | 5 | ***GSDMB ORMDL3*** *PGAP3* (*GSDMA*) | **CD4 CD8 CD19** CD14 **IL TR RE** | −0.98 | −0.92 | 0.02 | 0.01 | 2,4 |
| HD161 | 17 | 40.3 | 41.0 | 836 | 2 | *STAT3* | PLA | 0.67 | 0 | 0.10 | 1.00 | |
| HD164 | 18 | 67.4 | 67.6 | 1988 | 1 | CD226 | CD4 CD8 PLA | 0 | −0.86 | 1.00 | 0.01 | 2 |
| N_18_76 | 18 | 76.7 | 77.7 | 7292 | 1 | PQLC1 | PLA | −0.68 | 0 | 0.01 | 1.00 | |
| HD166 | 19 | 10.3 | 10.7 | 9232 | 1 | (*TYK2*) | CD14 | −0.44 | −0.09 | 0.10 | 0.10 | |
| HD168 | 19 | 47.1 | 47.4 | 581 | 2 | GNG8 | CD4 | 0 | −0.63 | 1.00 | 0.06 | |
| HD169 | 19 | 49.0 | 49.3 | 3128 | 1 | ***FUT2*** | **IL TR RE** | −0.95 | 0 | 0.01 | 1.00 | 4 |
| J_20_31 | 20 | 31.1 | 31.6 | 593 | 2 | COMMD7 | CD14 | 0 | 0.61 | 1.00 | 0.01 | |
| J_20_32 | 20 | 33.6 | 34.1 | 7 | 8 | UQCC1 | CD19 | −0.69 | 0 | 0.02 | 1.00 | 2 |
| | | | | 3369 | 1 | MMP24-AS1 | RE | −0.63 | −0.71 | 0.03 | 0.03 | |
| HD175 | 20 | 62.2 | 62.5 | 2322 | 1 | *LIME1* | CD4 CD19 | −0.86 | 0 | 0.01 | 1.00 | 2 |
| HD176 | 21 | 16.6 | 16.9 | 9578 | 1 | NRIP1 | CD4 | 0 | −0.69 | 1.00 | 0.02 | |
| HD180 | 22 | 21.7 | 22.1 | 2130 | 1 | **UBE2L3** | **CD4 CD8 CD19 CD14 CD15 IL TR RE** | 0.97 | 0.92 | 0.01 | 0.07 | 2,4 |
| N_22_41 | 22 | 41.4 | 42.4 | 2149 | 1 | EP300 | CD8 CD19 CD15 | 0 | 0.71 | 1.00 | 0.02 | |

Given are (i) the name and chromosomal coordinates of the corresponding loci (Locus, Chr, Beg, End) (GRCh37/hg19 in Mb), (ii) the identifier and total number of genes in the matching *cis*-acting regulatory module (cRM, Nr), (iii) the genes and tissues involved in matching DAP–EAP ($|\vartheta| > 0.6$) (bold when $|\vartheta| \geq 0.9$), (iv) the best $\vartheta$-values and corresponding empirical $p$ values obtained for CD and UC, respectively, and (vi) references reporting a link between one or more of the same genes and IBD on the basis of eQTL information. Genes that were resequenced are shown in italics. Genes that were resequenced despite $|\vartheta| \leq 0.6$ are bracketed, and the supporting references provided in "Ref". The higher number of matching DAP–EAP in this study when compared to Huang et al.[4] are primarily due to the fact that (i) we herein study 200 IBD risk loci (vs 97), and (ii) we increase the number of detected *cis*-eQTL approximately two-fold by correcting for hidden confounders using PCs

Although challenging, these numbers are potentially within reach of international consortia for several common diseases including IBD.

It is conceivable that the organ-specificity of nearly all complex diseases (such as the digestive tract for IBD), reflects tissue-specific perturbation of broadly expressed causative genes that may fulfill diverse functions in different organs. If this is true, coding variants may not be the appropriate substrate to perform burden tests, as these will affect the gene across all tissues. In such instances, the disruptive variants of interest may be those perturbing tissue-specific gene switches. Also, it has recently been proposed that the extreme polygenic nature of common complex diseases may reflect the trans-effects of a large proportion of regulatory variants active in a given cell type on a limited number of core genes via perturbation of highly connected gene networks[34]. Identifying rare regulatory variants is still challenging, however, as tissue-specific gene switches remain poorly cataloged, and the effect of variants on their function difficult to predict. The corresponding sequence space may also be limited in size, hence limiting power. Nevertheless, a reasonable start may be to re-sequence the regions surrounding common regulatory variants that have been fine-mapped at near single base pair resolution[4].

In conclusion, we hereby provide to the scientific community a collection of ~24,000 *cis*-eQTL in nine cell types that are highly relevant for the study of inflammatory and immune-mediated diseases, particularly of the intestinal tract. The CEDAR dataset advantageously complements existing eQTL datasets including GTEx[10,33]. We propose a paradigm to rationally organize *cis*-eQTL effects in co-regulated clusters or regulatory modules. We identify ~100 candidate causative genes in 63 out of 200 analyzed risk loci, on the basis of correlated DAP and EAP. We have developed a web-based browser to share the ensuing results with the scientific community (http://cedar-web.giga.ulg.ac.be). The CEDAR website will imminently be extended to accommodate additional common complex disease for which GWAS data are publicly available. We show that the corresponding candidate genes are enriched in causative genes, however, that case–control cohorts larger than those used in this study (12,000 individuals) are required to formally demonstrate causality by means of presently available burden tests.

## Methods

**Sample collection in the CEDAR cohort**. We collected peripheral blood as well as intestinal biopsies (ileum, transverse colon, rectum) from 323 healthy Europeans visiting the Academic Hospital of the University of Liège as part of a national screening campaign for colon cancer. Participants included 182 women and 141 men, averaging 56 years of age (range: 19-86). Enrolled individuals were not suffering any autoimmune or inflammatory disease and were not taking corticosteroids or non-steroid anti-inflammatory drugs (with the exception of low doses of aspirin to prevent thrombosis). We recorded birth date, weight, height, smoking history, declared ethnicity and hematological parameters (red blood cell count, platelet count, differential white blood cell count) for each individual. The experimental protocol was approved by the ethics committee of the University of Liège Academic Hospital. Informed consent was obtained prior to donation in agreement with the recommendations of the declaration of Helsinki for experiments involving human subjects. We refer to this cohort as CEDAR for Correlated Expression and Disease Association Research.

**SNP genotyping and imputation**. Total DNA was extracted from EDTA-collected peripheral blood using the MagAttract DNA blood Midi M48 Kit on a QIAcube robot (Qiagen). DNA concentrations were measured using the Quant-iT Picogreen ds DNA Reagents (Invitrogen). Individuals were genotyped for >700 K SNPs using Illumina's Human OmniExpress BeadChips, an iScan system and the Genome Studio software following the guidelines of the manufacturer. We eliminated variants with call rate ≤0.95, deviating from Hardy–Weinberg equilibrium ($p \leq 10^{-4}$), or which were monomorphic. We confirmed European ancestry of all individuals by PCA using the HapMap population as reference. Using the real genotypes of 629,570 quality-controlled autosomal SNPs as anchors, we used the Sanger Imputation Services with the UK10K + 1000 Genomes Phase 3 Haplotype panels (https://imputation.sanger.ac.uk)[35–37] to impute genotypes at autosomal variants in our population. We eliminated indels, SNPs with MAF ≤ 0.05, deviating from

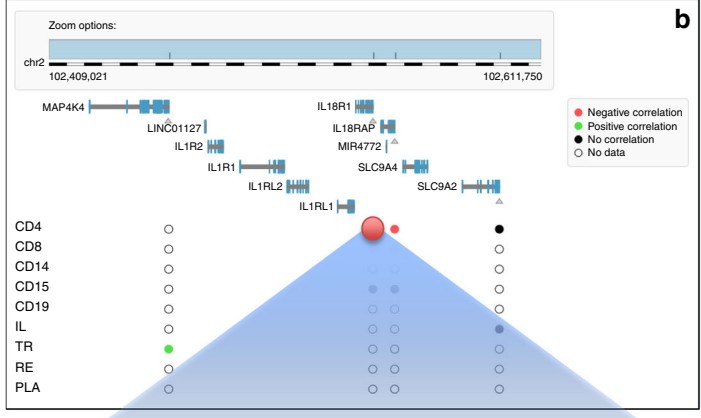

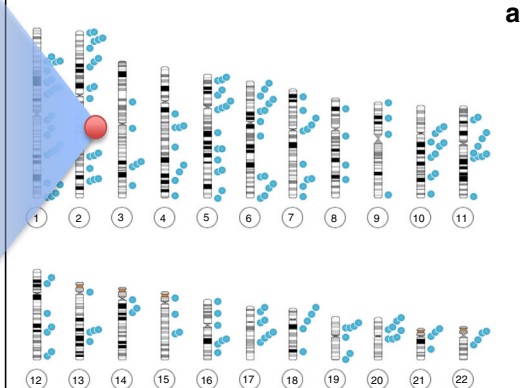

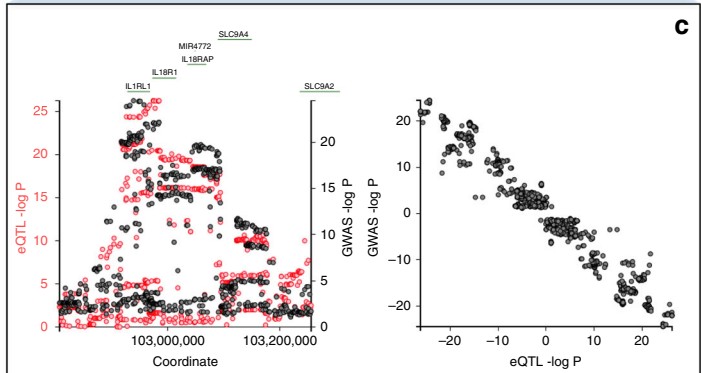

**Fig. 7** Screen shots of the CEDAR website, showing **a** known CD risk loci on the human karyotype, **b** a zoom in the HD35 risk locus showing the Refseq gene content and summarizing local CEDAR *cis*-eQTL data (white: no expression data, gray: expression data but no evidence for *cis*-eQTL, black: significant *cis*-eQTL but no correlation with DAP, red: significant *cis*-eQTL similar to DAP ($\vartheta < -0.60$), green: significant *cis*-eQTL similar to DAP ($\vartheta > 0.60$)), and **c** a zoom in the DAP for Crohn's disease (black) and EAP for *IL18R1* (red), as well as the signed correlation between DAP and EAP

Hardy-Weinberg equilibrium ($p \leq 10^{-3}$), and with low imputation quality (INFO ≤ 0.4), leaving 6,019,462 high quality SNPs for eQTL analysis.

**Transcriptome analysis.** Blood samples were kept on ice and treated within 1 h after collection as follows. EDTA-collected blood was layered on Ficoll-Paque PLUS (GE Healthcare) to isolate peripheral blood mononuclear cells by density gradient centrifugation. CD4+ T lymphocytes, CD8+ T lymphocytes, CD19+ B lymphocytes, CD14+ monocytes, and CD15+ granulocytes were isolated by positive selection using the MACS technology (Miltenyi Biotec). To isolate platelets, blood collected on acid-citrate-dextrose (ACD) anticoagulant was centrifuged at 150 g for 10 min. The platelet rich plasma (PRP) was collected, diluted twofold in ACD buffer and centrifuged at 800 × g for 10 min. The platelet pellet was resuspended in MACS buffer (Miltenyi Biotec) and platelets purified by negative selection using CD45 microbeads (Miltenyi Biotec). Intestinal biopsies were flash frozen in liquid nitrogen immediately after collection and kept at −80 °C until RNA extraction. Total RNA was extracted from the purified leucocyte populations and intestinal biopsies using the AllPrep Micro Kit and a QIAcube robot (Qiagen). For platelets, total RNA was extracted manually with the RNeasy Mini Kit (Qiagen). Whole genome expression data were generated using HT-12 Expression Beadchips following the instructions of the manufacturer (Illumina). Technical outliers were removed using controls recommended by Illumina and the Lumi package[38]. We kept 29,464/47,323 autosomal probes (corresponding to 19,731 genes) mapped by Re-Annotator[39] to a single gene body with ≤2 mismatches and not spanning known variants with MAF>0.05. Within cell types, we only considered probes (i.e., "usable" probes) with detection $p$ value ≤0.05 in ≥25% of the samples. Fluorescence intensities were Log$_2$ transformed and Robust Spline Normalized (RSN) with Lumi[38]. Normalized expression data were corrected for sex, age, smoking status and Sentrix Id using ComBat from the SVA R library[40]. We further corrected the ensuing residuals within tissue for the number of Principal Components (PC) that maximized the number of *cis*-eQTL with $p \leq 10^{-6}$ [41]. Supplementary Table 2 summarizes the number of usable samples, probes and PC for each tissue type.

**cis-eQTL analysis.** *cis*-eQTL analyses were conducted with PLINK and using the expression levels precorrected for fixed effects and PC as described above (http://pngu.mgh.harvard.edu/purcell/plink/)[42]. Analyses were conducted under an

additive model, i.e., assuming that the average expression level of heterozygotes is at the midpoint between alternate homozygotes. To identify *cis*-eQTL we tested all SNPs in a 2 Mb window centered around the probe (if "usable"). $P$ values for individual SNPs were corrected for the multiple testing within the window by permutation (10,000 permutations). For each probe–tissue combination we kept the best (corrected) $p$ value. Within each individual cell type, the ensuing list of corrected $p$ values was used to compute the corresponding false discovery rates (FDR or $q$ value). Supplementary Table 3 reports the number of *cis*-eQTL found in the nine analyzed cell types for different FDR thresholds (see also Supplementary Fig. 9).

**Comparing EAP with $\vartheta$ to identify *cis*-regulatory modules.** If the transcript levels of a given gene are influenced by the same regulatory variants (one or several) in two tissues, the corresponding EAP (i.e., the $-\log(p)$ values of association for the SNPs surrounding the gene) are expected to be similar. Likewise, if the transcript levels of different genes are influenced by the same regulatory variants in the same or in different tissues, the corresponding EAP are expected to be similar (cf. main text, Fig. 1). We devised a metric, $\vartheta$, to quantify the similarity between EAP. If two EAP are similar, one can expect the corresponding $-\log(p)$ values to be positively correlated. One particularly wants the EAP peaks, i.e., the highest $-\log(p)$ values, to coincide in order to be convinced that the corresponding *cis*-eQTL are driven by the same regulatory variants. To quantify the similarity between EAP while emphasizing the peaks, we developed a weighted correlation. Imagine two vectors **X** and **Y** of $-\log(p)$ values for $n$ SNPs surrounding the gene(s) of interest. Using the same nomenclature as in Fig. 1a, **X** could correspond to gene A in tissue 1, and **Y** to gene A in tissue 2, or **X** could correspond to gene A in tissue 1, and **Y** to gene B in tissue 2. We only consider for analysis, SNPs within 1 Mb of either gene (probe) and for which $x_i$ and/or $y_i$ is superior to 1.3 (i.e., $p$ value <0.05) hence informative for at least one of the two *cis*-eQTL. Indeed, the majority of variants with $-\log(p)$ <1.3 ($p > 0.05$) for both EAP are by definition not associated with either trait. There is therefore no reason to expect that they could contribute useful information to the correlation metric: their ranking in terms of $-\log(p)$ values becomes more and more random as the $-\log(p)$ decreases. We define the weight to be given to each SNP

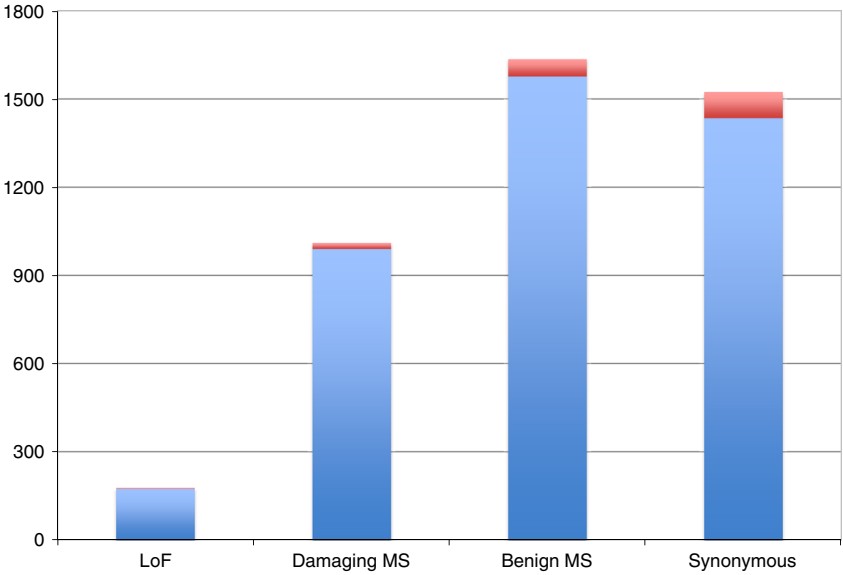

**Fig. 8** Variants detected by sequencing the coding exons of 45 candidate genes. Variants are sorted in LoF (loss-of-function, i.e., stop gain, frame-shift, splice site), Damaging MS (missense variants considered as damaging by SIFT[5] and damaging or possibly damaging by Polyphen-2[6]), Benign MS (other missense variants), and Synonymous. Blue: variants with MAF < 0.005, Red: variants with MAF ≥ 0.005

in the correlation as:

$$w_i = \left( \text{MAX} \left( \frac{x_i}{x_{\text{MAX}}}, \frac{y_i}{y_{\text{MAX}}} \right) \right)^p$$

The larger $p$, the more weight is given to the top SNPs. In this work, $p$ was set at one.

The weighted correlation between the two EAP, $r_w$, is then computed as:

$$r_w = \frac{1}{\sum_{i=1}^n w_i} \sum_{i=1}^n w_i \left( \frac{x_i - \overline{x_w}}{\sigma_x^w} \right) \left( \frac{y_i - \overline{y_w}}{\sigma_y^w} \right)$$

in which

$$\overline{x_w} = \frac{\sum_{i=1}^n w_i \times x_i}{\sum_{i=1}^n w_i}$$

$$\overline{y_w} = \frac{\sum_{i=1}^n w_i \times y_i}{\sum_{i=1}^n w_i}$$

$$\sigma_x^w = \sqrt{\frac{\sum_{i=1}^n w_i \times (x_i - \overline{x_w})^2}{\sum_{i=1}^n w_i}}$$

$$\sigma_y^w = \sqrt{\frac{\sum_{i=1}^n w_i \times (y_i - \overline{y_w})^2}{\sum_{i=1}^n w_i}}$$

The larger $r_w$, the larger the similarity between the EAP, particularly for their respective peak SNPs.

$r_w$ ignores an important source of information. If two EAP are driven by the same regulatory variant, there should be consistency in the signs of the effects across SNPs in the region. We will refer to the effect of the "reference" allele of SNP $i$ on the expression levels for the first and second cis-eQTL as $\beta_i^X$ and $\beta_i^Y$. If the reference allele of the regulatory variant increases expression for both cis-eQTL, the $\beta_i^X$ and $\beta_i^Y$'s for a SNPs in LD with the regulatory variant are expected to have the same sign (positive or negative depending on the sign of D for the considered SNP). If the reference allele of the regulatory variant increases expression for one cis-eQTL and decreases expression for the other, the $\beta_i^X$ and $\beta_i^Y$'s for a SNPs in LD with the regulatory variant are expected to have opposite sign. We used this notion to develop a weighted and signed measure of correlation, $r_{ws}$. The approach was the same as for $r_w$, except that the values of $y_i$ were multiplied by $-1$ if the signs of $\beta_i^X$

and $\beta_i^Y$ were opposite. $r_{ws}$ is expected to be positive if the regulatory variant affects the expression of both cis-eQTL in the same direction and negative otherwise.

We finally combined $r_w$ and $r_{ws}$ in a single score referred to as $\vartheta$, as follows:

$$\vartheta = \frac{r_{ws}}{1 + e^{-k(r_w - T)}}$$

$\vartheta$ penalizes $r_{ws}$ as a function of the value of $r_w$. The aim is to avoid considering EAP pairs with strong but negative $r_w$ (which is often the case when the two EAP are driven by very distinct variants). The link function is a sigmoid-shaped logistic function with $k$ as steepness parameter and $T$ as sigmoid mid-point. In this work, we used a value of $k$ of 30, and a value of $T$ of 0.3 (Supplementary Fig. 10).

We first evaluated the distribution of $\vartheta$ for pairs of EAP driven by the same regulatory variants by studying 4,693 significant cis-eQTL (FDR < 0.05). For these, we repeatedly (100x) split our CEDAR population in two halves, performed the cis-eQTL analysis separately on both halves and computed $\vartheta$ for the ensuing EAP pairs. Supplementary Fig. 1 is showing the obtained results.

We then evaluated the distribution of $\vartheta$ for pairs of EAP driven by distinct regulatory variants in the same chromosomal region as follows. We considered 1207 significant cis-eQTL (mapping to the 200 IBD risk loci described above). For each one of these, we generated a set of 100 "matching" cis-eQTL effects in silico, sequentially considering 100 randomly selected SNPs (from the same locus) as causal. The in silico cis-eQTL were designed such that they would explain the same fraction of expression variance as the corresponding real cis-eQTL detected with PLINK (cfr. above). When performing cis-eQTL analysis under an additive model, PLINK estimates $\beta_0$ (i.e., the intercept), and $\beta_1$ (i.e., the slope of the regression), including for the top SNP. Assume that the expression level of the studied gene, Z, for individual $i$ is $z_i$. Assume that the sample comprises $n_T$ individuals in total, of which $n_{11}$ are of genotype "11", $n_{12}$ of genotype "12", and $n_{22}$ of genotype "22", for the top cis-eQTL SNP. The total expression variance for gene Z equals:

$$\sigma_T^2 = \frac{\sum_{i=1}^{n_T} (z_i - \overline{z_T})^2}{n_T - 1}$$

The variance in expression level due to the cis-eQTL equals:

$$\sigma_{\text{eQTL}}^2 = \frac{n_{11}(\beta_0 - \overline{z_T})^2 + n_{12}(\beta_0 + \beta_1 - \overline{z_T})^2 + n_{22}(\beta_0 + 2\beta_1 - \overline{z_T})^2}{n_T}$$

The heritability of expression due to the cis-eQTL, i.e., the fraction of the expression variance that is due to the cis-eQTL is therefore:

$$h_{\text{eQTL}}^2 = \frac{\sigma_{\text{eQTL}}^2}{\sigma_T^2}$$

To simulate cis-eQTL explaining the same $h_{\text{eQTL}}^2$ as the real eQTL in the CEDAR dataset, we sequentially considered all SNPs in the region. Each one of these SNPs would be characterized by $n_{11}$ individuals of genotype "11", $n_{12}$ of

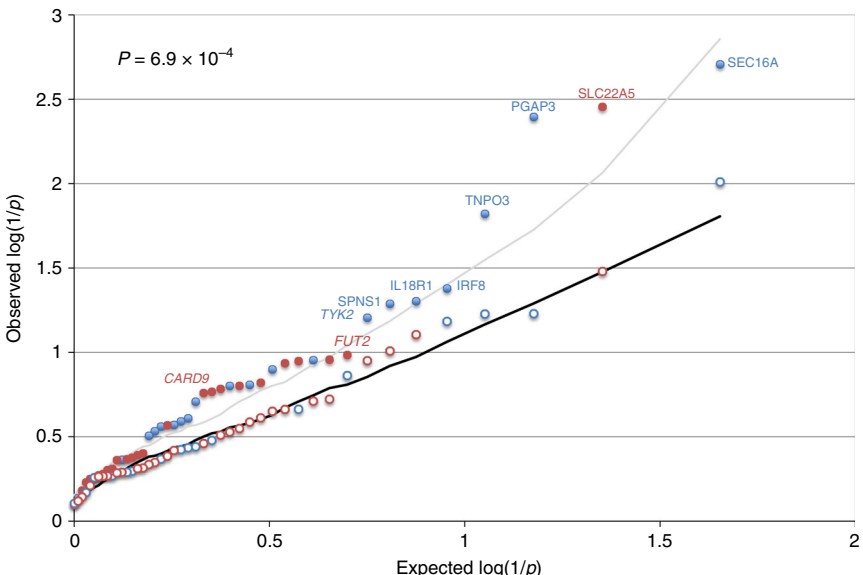

**Fig. 9** QQ-plot for the gene-based burden test. Ranked $\log(1/p)$ values obtained when considering LoF and damaging variants (full circles), or synonymous variants (empty circles). The circles are labeled in blue when the best $p$ value for that gene is obtained with CAST, in red when the best $p$ value is obtained with SKAT. The black line corresponds to the median $\log(1/p)$ value obtained (for the corresponding rank) using the same approach on permuted data (LoF and damaging variants). The gray line marks the upper limit of the 95% confidence band. The name of the genes with nominal $p$ value ≤0.05 are given. Known causative genes are italicized. The inset $p$ value corresponds to the significance of the upwards shift in $\log(1/p)$ values estimated by permutation

genotype "12", and $n_{22}$ of genotype "22", for a total of $n_T$ genotyped individuals. We would arbitrarily set $\overline{z_{11}}$, $\overline{z_{12}}$, and $\overline{z_{22}}$ at −1, 0, and +1. As a consequence, the variance due to this *cis*-eQTL equals:

$$\sigma^2_{\text{eQTL}} = \frac{n_{11}(-1 - \overline{z_T})^2 + n_{12}(0 - \overline{z_T})^2 + n_{22}(1 - \overline{z_T})^2}{n_T}$$

in which $\overline{z_T} = (n_{22} - n_{11})/n_T$.

Knowing $\sigma^2_{\text{eQTL}}$ and $h^2_{\text{eQTL}}$, and knowing that

$$h^2_{\text{eQTL}} = \frac{\sigma^2_{\text{eQTL}}}{\sigma^2_{\text{eQTL}} + \sigma^2_{\text{RES}}}$$

the residual variance $\sigma^2_{\text{RES}}$ can be computed as

$$\sigma^2_{\text{RES}} = \sigma^2_{\text{eQTL}}\left(\frac{1}{h^2_{\text{eQTL}}} - 1\right)$$

Individual expression data for the corresponding *cis*-eQTL (for all individuals of the CEDAR data set) were hence sampled from the normal distribution

$$z_i \sim N\left(\overline{z_{xx}}, \sigma^2_{\text{RES}}\right)$$

where $\overline{z_{xx}}$ is −1, 0, or +1 depending on the genotype of the individual (11, 12, or 22). We then performed *cis*-eQTL on the corresponding data set using PLINK, generating an in silico EAP. Real and in silico EAP were then compared using $\vartheta$. Supplementary Fig. 1 shows the corresponding distribution of $\vartheta$ values for EAP driven by distinct regulatory variants.

The corresponding distributions of $\vartheta$ under $H_1$ and $H_0$ (Supplementary Fig. 1) show that $\vartheta$ discriminates very effectively between $H_1$ and $H_0$ especially for the most significant *cis*-eQTL. We chose a threshold of $|\vartheta| > 0.6$ to cluster EAP in *cis*-acting regulatory elements or cRM (Fig. 2). In the experiment described above, this would yield a false positive rate of 0.05, and a false negative rate of 0.23. Clusters were visually examined as show in Supplementary Fig. 2. Twenty-nine edges connecting otherwise unlinked and yet tight clusters were manually removed.

**Testing for an excess sharing of cRM between cell types.** Assume that cell type 1 is part of $n_{1T}$ cRM, including $n_{11}$ private cRM, $n_{12}$ cRM shared with cell type 2, $n_{13}$ cRM shared with cell type 3, …, and $n_{19}$ cRM shared with cell type 9. Note that $\sum_{i=1}^{9} n_{1i} \geq n_{1T}$, because cRM may include more than two cell types. Assume that $n_{1S} = \sum_{i=1}^{9} n_{1i}$ is the sum of pair-wise sharing events for cell type 1. We computed, for each cell type $i \neq 1$, the probability to observe $\geq n_{1i}$ sharing events with cell type 1

assuming that the expected number (under the hypothesis of random assortment) is

$$n_{1S} \times \frac{n_{iT}}{\sum_{j \neq 1}^{9} n_{jT}}$$

Pair-wise sharing events between tissue 1 and the eight other tissues were generated in silico under this model of random assortment (5000 simulations). The $p$ value for $n_{1i}$ was computed as the proportion of simulations that would yield values that would be as large or larger than $n_{1i}$. The same approach was used for the nine cell types. Thus, two $p$ values of enrichment are obtained for each pair of cell types $i$ and $j$, one using $i$ as reference cell type, and the other using $j$ as reference cell type. As can be seen from Fig. 5, the corresponding pairs of $p$ values were always perfectly consistent.

We performed eight distinct analyses. In the first analysis, we only considered cRM involving no more than two tissues (i.e., unique for specific pairs of cell types). In subsequent analyses, we progressively included cRM with no more than three, four, …, and nine cell types.

**Comparing EAP and DAP using $\vartheta$.** The approach used to cluster EAP in cRM was also used to assign DAP for Inflammatory Bowel Disease (IBD) to EAP-defined cRM. We studied 200 IBD risk loci identified in recent GWAS meta-analyses[2,3]. The limits of the corresponding risk loci were as defined in the corresponding publications. We measured the similarity between DAP and EAP using the $\vartheta$ metric for all *cis*-eQTL mapping to the corresponding intervals (i.e., for all *cis*-eQTL for which the top SNP mapped within the interval). To compute the correlations between DAP and EAP we used all SNPs mapping to the disease interval with –log $(p)$ value ≥1.3 either for DAP, EAP or both.

In addition to computing $\vartheta$ as described in section 5, we computed an empirical $p$ value for $\vartheta$ using the approach (based on in silico generated *cis*-eQTL) described above to generate the locus-specific distribution of $\vartheta$ values for EAP driven by distinct regulatory variants. From this distribution, one can deduce the probability that a randomly generated EAP (explaining as much variance as the real tested EAP) and the DAP would by chance have a $|\vartheta|$ value that is as high or higher than the real EAP. The corresponding empirical $p$ value accounts for the local LD structure between SNPs.

**Evaluating the enrichment of DAP–EAP matching.** To evaluate whether DAP matched EAP more often than expected by chance alone, we analyzed 97 IBD risk loci interrogated by the Immunochip, (i) in order to allow for convenient comparison with Huang et al.[4], and (ii) because we needed extensively QC genotypes for the IIBDGC data to perform the enrichment analysis with the $\vartheta$-based method (see hereafter). Within these 97 IBD risk loci, we focused on 63 regions affecting CD[4], encompassing at least one significant eQTL, and for which the lead CD-associated SNP had MAF > 0.05. Indeed, eQTL analyses in the CEDAR dataset

were restricted to SNPs with MAF > 0.05 (see above). We used three methods to evaluate whether the observed number of DAP–EAP matches were higher than expected by chance alone: naïve, frequentist and $\vartheta$-based. Analyses were performed separately for the nine cell types.

In the "naïve" approach, DAP and EAP were assumed to match if the corresponding lead SNPs were in LD with $r^2 \geq 0.8$. This would yield $n_N \leq 63$ risk loci for which the DAP would match at least one EAP. To measure the statistical significance of $n_N$, we sampled a SNP (MAF > 0.05) at random in each of the 63 risk loci, and counted the number of loci with at least one matching EAP. This "simulation" was repeated 1,000 times. The significance of $n_N$ was measured as the proportion of simulations that would yield $\geq n_N$ matches.

The frequentist approach used the method described by Nica et al.[11]. DAP and EAP were assumed to match if fitting the disease-associated lead SNP in the eQTL analysis caused a larger drop in $-\log(p)$ than 95% of the SNPs with MAF > 0.05 in the analyzed risk locus. This would yield $n_F \leq 63$ risk loci for which the DAP would match at least one EAP. To measure the statistical significance of $n_F$, we sampled a SNP (MAF > 0.05) at random in each of the 63 risk loci, and counted the number of loci with at least one matching EAP. This "simulation" was repeated 1000 times. The significance of $n_F$ was measured as the proportion of simulations that would yield $\geq n_F$ matches.

Finally, we used our $\vartheta$-based approach in which DAP and EAP were assumed to match if $|\vartheta| > 0.6$. This would yield $n_\vartheta \leq 63$ risk loci for which the DAP would match at least one EAP. To measure the statistical significance of $n_\vartheta$ we sampled a SNP (MAF > 0.05) at random in each of the 63 risk loci, and generated a DAP assuming that the corresponding SNPs were causal as follows.

Assume a cohort with $n_1$ cases and $n_2$ controls (for instance, the IIBDGC cohort). Assume a SNP with an allelic frequency of $p$ in the cases + controls, an allelic frequency of $(p + d)$ in cases and $(p + \delta)$ in controls.

One can easily show that:

$$\delta = -d \frac{n_1}{n_2} \tag{1}$$

The odds ratio (OR) for that SNP equals:

$$\mathrm{OR} = \frac{(p + d)(1 - p - \delta)}{(p + \delta)(1 - p - d)}$$

The ratio between the between-cohort (i.e., cases and controls) variance versus within-cohort variance (corresponding to an $F$ test) can be shown to equal:

$$F = \frac{d^2 \left(1 + \frac{n_1}{n_2}\right)}{\left(1 + \frac{n_2}{n_1}\right)(p - p^2) - d^2 \left(1 + \frac{n_1}{n_2}\right)}$$

If we fix $F$ based on the real top SNP in the IIBDGC data in a given GWAS identified risk loci, we can determine $d$ (and hence $\delta$ using Equation 1) for the randomly selected SNP (that will become an "in silico causative variant") with allelic frequency in (cases + controls) of $p$ (different from the real top SNP), by solving

$$d = \frac{-\beta \pm \sqrt{\beta^2 - 4\alpha\gamma}}{2\alpha}$$

where

$$\alpha = \left(1 + \frac{n_1}{n_2}\right)(1 + F)$$

$$\beta = 0$$

$$\gamma = -\left(p - p^2\right)\left(1 + \frac{n_2}{n_1}\right)F$$

Once we know $(p + d)$ (i.e., the frequency of the SNP in cases), and hence $(p + \delta)$ (i.e., the frequency of the SNP in controls), we can use Hardy–Weinberg to determine the frequency of the three genotypes in cases ($p_{AA}^{IBD}, p_{AB}^{IBD}, p_{BB}^{IBD}$) and controls ($p_{AA}^{CTR}, p_{AB}^{CTR}, p_{BB}^{CTR}$). We then create an in silico case–control cohort by sampling (with replacement) $n_1 \times p_{AA}^{IBD}$ AA cases, $n_1 \times p_{AB}^{IBD}$ AB cases, …, and $n_2 \times p_{BB}^{CTR}$ BB controls from the individuals of the IIBDGC (without discriminating real case and control status). Association analysis of the corresponding data set in the chromosome region of interest generates DAP with max $-\log(p)$ value similar to the real DAP. This "simulation" was repeated 1000 times. The significance of $n_\vartheta$ was measured as the proportion of simulations that would yield $\geq n_\vartheta$ matches.

**Targeted exon resequencing in CD cases and controls**. Genes for which EAP match the DAP tightly (high $|\vartheta|$ values) are strong candidate causal genes for the studied disease. In the case of IBD, we identified ~100 such genes (Table 1). Ultimate proof of causality can be obtained by demonstrating a differential burden of rare disruptive variants in cases and controls. Burden tests preferably focus on coding gene segments, in which disruptive variants are most effectively recognized. Analyses are restricted to rare variants to ensure independence from the GWAS signals.

To perform burden tests, we collected DNA samples from 7323 Crohn Disease (CD) cases and 6342 controls of European descent in France (cases: 1899—ctrls: 1731), the Netherlands (2002–1923) and, Belgium (3422–2688). The study protocols were approved by the institutional review board at each center involved with recruitment. Informed consent and permission to share the data were obtained from all subjects, in compliance with the guidelines specified by the recruiting center's institutional review board.

During the course of this project, we selected 45 genes with high $|\vartheta|$ values for resequencing (Table 1). We designed primers to amplify all corresponding coding exons plus exon-intron boundaries corresponding to all transcripts reported in the CCDS release 15[43] (Supplementary Data 8). Following Momozawa et al.[24], the primers were merged in five pools to perform a first round of PCR amplification (25 cycles). We then added 8-bp barcodes and common adapters (for sequencing) to all PCR products by performing a second round of PCR amplification (4 cycles) using primers targeting shared 5′ overhangs introduced during the first PCR. The ensuing libraries were purified, quality controlled and sequenced (2 × 150-bp paired-end reads) on a HiSeq 2500 (Illumina) instrument. Sequence reads were sorted by individual using the barcodes, aligned to the human reference sequence (hg19) with the Burrows–Wheeler Aligner (ver. 0.7.12)[44], and further processed using Genome Analysis Toolkit (GATK, ver. 3.2-2)[45]. We only considered individuals for further analyses if ≥95% of the target regions was covered by ≥20 sequence reads. Average sequence depth across individuals and target regions was 1060. We called variants for each individual separately using the UnifiedGenotyper and HaplotypeCaller of GATK, as well as VCMM (ver. 1.0.2)[46], and listed all variants detected by either method. Genotypes for all individuals were determined for each variant based on the ratio of reference and alternative alleles amongst sequence reads as determined by Samtools[47]. Individuals were labeled homozygote reference, heterozygote, or homozygote derived when the alternative allele frequency was between 0 and 0.15, between 0.25 and 0.75, and between 0.85 and 1, respectively. If the alternative allele frequency was outside these ranges or a variant position was covered with <20 sequencing reads, the genotype was considered missing. We excluded variants with call rates <95% or variants that were not in Hardy-Weinberg equilibrium ($P < 1 \times 10^{-6}$). We excluded 281 individuals with ≥2 minor alleles at 23 variants selected to have a MAF ≤ 0.01 in non-Finnish Europeans and ≥0.10 in Africans or East-Asians in the Exome Aggregation Consortium[27].

In the end, we used 6597 cases and 5502 controls for further analyses, while 98.5% of the target regions on average was covered with 20 or more sequence reads.

**Gene-based burden test**. We first used SIFT[25] and Polyphen-2[26] to sort the 4175 variants identified by sequencing in four categories: (i) loss-of-function (LoF) or severe, corresponding to stop gain, stop loss, frameshift and splice-site variants, (ii) damaging, corresponding to missense variants predicted by SIFT to be damaging and Polyphen-2 to be possibly or probably damaging, (iii) benign, corresponding to the other missense variants, and (iv) synonymous. We performed the burden test using the LoF plus damaging variants, and used the synonymous variants as controls. We only considered variants with MAF (computed for the entire data set, i.e., cases plus controls) ≤0.005. We indeed showed in a previous fine-mapping study that all reported independent effects were driven by variants with MAF ≥0.01[4]. By doing so we ensure that the signals of the burden test are independent of previously reported association signals. Thus, 174 LoF, 991 damaging, and 1434 synonymous were ultimately used to perform burden tests.

Burden tests come in two main flavors. In the first, one assumes that disruptive variants will be enriched in either cases (i.e., disruptive variants increase risk) or in controls (i.e., disruptive variance decrease risk). In the second, one assumes that—for a given gene—some disruptive variants will be enriched in cases, while other may be enriched in controls (Supplementary Fig. 11). The first was implemented using CAST[28]. To increase power, we exploited the DAP–EAP information to perform one-sided (rather than two-sided) tests. When $\vartheta < 0$, we tested for an enrichment of disruptive variants in cases; when $\vartheta > 0$, for an enrichment of disruptive variants in controls. $P$ values were computed by phenotype permutation, i.e., shuffling case–control status. When applying this test on a gene-by-gene basis using synonymous variants (MAF > 0.005), the distribution of $p$ values (QQ-plot) indicated that the CAST test was conservative ($\lambda_{GC} = 0.51$) (Supplementary Fig. 12). The second kind of burden test was implemented with SKAT[29]. It is noteworthy that SKAT ignores information from singletons (Supplementary Fig. 11). Just as for CAST, $p$ values were computed by phenotype permutation, i.e., shuffling case–control status. When applying this test on a gene-by-gene basis using synonymous variants (MAF < 0.005), the distribution of $p$ values (QQ-plot) indicated that the SKAT test is too permissive ($\lambda_{GC} = 1.73$) (Supplementary Fig. 12). Consequently, gene-based $p$ values obtained with SKAT were systematically GC corrected using this value of $\lambda_{GC}$. We performed the two kinds of

analyses for each gene, as one doesn't a priori know what hypothesis will match the reality best for a given gene.

We also extracted information from the distribution of $p$ values (or $-\log(p)$ values) across the 45 analyzed genes. Even if individual genes do not yield $-\log(p)$ values that exceed the significance threshold (accounting for the number of analyzed genes and tests performed), the distribution of $-\log(p)$ values may significantly depart from expectations, indicating that the analyzed genes include at least some causative genes. This was done by taking for each gene, the best $p$ value (whether obtained with CAST or SKAT) and then rank the genes by corresponding $-\log(p)$ value. The same was done for $10^5$ phenotype permutations, allowing us to examine the distribution of $-\log(p)$ values for given ranks and compute the corresponding medians and limits of the 95% confidence band, as well as to compute the probability that $-2\sum_{i=1}^{45}\ln(p_i)$ (Fisher's equation to combine $p$ values) equals or exceeds the observed. Our results show that there is a significant departure from expectation when analyzing the damaging variants ($p = 6.9 \times 10^{-4}$) but not when analyzing the synonymous variants ($p = 0.66$) supporting the presence of genuine causative genes amongst the analyzed list.

**cRM-based burden test**. The enrichment of multi-genic cRM in IBD risk loci suggests that risk loci may have more than one causative gene belonging to the same cRM. To capitalize on this hypothesis, we developed a cRM-based burden test. Gene-specific $p$ values were combined within cRM using Fisher's method. For each gene, we considered the best $p$ value whether obtained with CAST or SKAT. Statistical significance was evaluated by phenotype permutation exactly as described for the gene-based burden test. By doing so we observed a departure from expectation when using the damaging variants ($p = 2.3 \times 10^{-3}$), but not when using the synonymous variants ($p = 0.72$).

**Orthogonal tests for age-of-onset and familiality**. It is commonly assumed that the heritability for common complex diseases is higher in familial and early onset cases[31]. To extract the corresponding information from our data in a manner that would be orthogonal to the gene- and module-based tests described above (i.e., the information about age-of-onset and familiality would be independent of these burden tests), we devised the following approach.

For age-of-onset, we summed the age-of-onset of the $n_C$ cases carrying rare disruptive variants for the gene of interest. We then computed the probability that the sum of the age-of-onset of $n_C$ randomly chosen cases was as different from the mean of age-of-onset as the observed one, yielding a gene-specific two-sided $p_{SKAT}$ value. In addition, we used the eQTL information to generate gene-specific one-sided $p_{CAST}$ values, corresponding to the probability that the sum of the age-of-onset of $n_C$ randomly chosen cases was as low or lower than the observed one (for genes for which decrease in expression level as associated with increased risk), or to the probability that the sum of the age-of-onset of $n_C$ randomly chosen cases was as high or higher than the observed one (for genes for which increase in expression level as associated with increased risk). These age-of-onset $p$ values were then combined with the corresponding $p$ values from the burden test (CAST with CAST, SKAT with SKAT) using Fisher's method.

For familiality, we determined what fraction of the $n_C$ cases carrying rare disruptive variants for the gene of interest were familial (affected first degree relative). We then computed the probability that the fraction of familial cases amongst $n_C$ randomly chosen cases was as different from the overall proportion of familial cases, yielding a gene-specific two-sided $p_{SKAT}$ value. In addition, we used the eQTL information to generate gene-specific one-sided $p_{CAST}$ values, corresponding to the probability that the fraction of familial cases amongst $n_C$ randomly chosen cases was as high or higher than the observed one (for genes for which decrease in expression level as associated with increased risk), or to the probability that the sum of the age-of-onset of $n_C$ randomly chosen was as low or lower than the observed one (for genes for which increase in expression level as associated with increased risk). These familial $p$ values were then combined with the corresponding $p$ values from the burden test (CAST with CAST, SKAT with SKAT) using Fisher's method.

**Data availability**. The complete CEDAR eQTL dataset can be downloaded from the Array Express website (https://www.ebi.ac.uk/arrayexpress/), accession numbers E-MTAB-6666 (genotypes) and E-MTAB-6667 (expression data). The data, preprocessed as described in Methods, can be downloaded from the CEDAR website (http://cedar-web.giga.ulg.ac.be).

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

## Acknowledgements

This work was supported by grants to Michel Georges from WELBIO (CAUSIBD), BELSPO (BeMGI), and Horizon 2020 (SYSCID). Computational resources at ULg have been provided by GIGA and the Consortium des Équipements de Calcul Intensif (CÉCI), funded by the Fonds de la Recherche Scientifique de Belgique (F.R.S.-FNRS) under Grant No. 2.5020.11. This work was conducted as part of the BioBank Japan Project supported by the Japan Agency for Medical Research and Development and by the Ministry of Education, Culture, Sports, Sciences and Technology of the Japanese government. The work of D.A. and I.A. was supported by Russian Ministry of Science and Education under 5-100 Excellence Programme. R.K. Weersma is supported by a VIDI grant (016.136.308) from the Netherlands Organisation for Scientific Research (NWO). DNA samples from the Dutch IBD cohort have been collected within the Parelsnoer Institute Project. This nationwide Parelsnoer Institute project is part of and funded by the Netherlands Federation of University Medical Centres and has received initial funding from the Dutch Government (from 2007 to 2011). The Parelsnoer Institute currently facilitates the uniform nationwide collection of information on and biomaterials of thirteen other diseases. We are grateful to N. Hakozaki, H. Iijima, N. Maki, and other staff of the Laboratory for Genotyping Development, RIKEN Center for the Integrative Medical Sciences. We thank Wouter Coppieters and the other members of the GIGA genomics platform for their support.

## Author contributions

Y.M., J.D., and M.G. conceived experiments, generated data, analyzed data and wrote the manuscript. E.T., V.D., S.R., B.C., F.C., E.D., M.E., A.-S.G., C.L., R.M., M.M., and C.O. generated and analyzed data. I.A., D.A., Y.A., and M.G. conceived and generated the CEDAR website. L.A., G.B., F.H., M.L., B.O., M.J.P., A.E.v.d.M.-d.J., J.J.v.d.W., M.C.V., M.L., J.-P.H., R.K.W., M.D.V., D.F., S.V., M.K., and E.L. collected and provided samples. The IIBDGC provided association *p*-values in the 201 IBD risk loci (DAPs).

## Additional information

**Competing interests:** The authors declare no competing interests.

Yukihide Momozawa[1,2], Julia Dmitrieva[1], Emilie Théâtre[1], Valérie Deffontaine[1], Souad Rahmouni[1], Benoît Charloteaux[1], François Crins[1], Elisa Docampo[1], Mahmoud Elansary[1], Ann-Stephan Gori[1], Christelle Lecut[3], Rob Mariman[1], Myriam Mni[1], Cécile Oury[3], Ilya Altukhov[4], Dmitry Alexeev[5], Yuri Aulchenko[6,7,8], Leila Amininejad[9], Gerd Bouma[10], Frank Hoentjen[11], Mark Löwenberg[12], Bas Oldenburg[13], Marieke J. Pierik[14], Andrea E. vander Meulen-de Jong[15], C. Janneke van der Woude[16], Marijn C. Visschedijk[17], The International IBD Genetics Consortium, Mark Lathrop[18], Jean-Pierre Hugot[19], Rinse K. Weersma[17], Martine De Vos[20], Denis Franchimont[9], Severine Vermeire[21], Michiaki Kubo[2], Edouard Louis[22] & Michel Georges[1]

[1]Unit of Animal Genomics, WELBIO, GIGA-R & Faculty of Veterinary Medicine, University of Liège (B34), 1 Avenue de l'Hôpital, Liège 4000, Belgium. [2]Laboratory for Genotyping Development, RIKEN Center for Integrative Medical Science, 1-7-22, Suehiro-cho, Tsurumi-ku, Yokohama, Kanagawa 230-0045, Japan. [3]Laboratory of Thrombosis and Hemostasis, GIGA-R, University of Liège (B34), 1 Avenue de l'Hôpital, 4000 Liège, Belgium. [4]Moscow Institute of Physics and Technology, Institutskiy Pereulok 9, Dolgoprudny 141700, Russian Federation. [5]Novosibirsk State University, Pirogova ave. 2, Novosibirsk 630090, Russian Federation. [6]PolyOmica, Het Vlaggeschip 61, 's-Hertogenbosch 5237 PA, The Netherlands. [7]Institute of Cytology and Genetics SD RAS, Lavrentyeva ave. 10, 630090 Novosibirsk, Russia. [8]Centre for Global Health Research,

Usher Institute of Population Health Sciences and Informatics, University of Edinburgh, Teviot Place, Edinburgh EH8 9AG, UK. [9]Gastroentérologie Médicale, Faculté de Médicine, Université Libre de Bruxelles, Route de Lennik 808, Anderlecht 1070, Belgium. [10]Department of Gastroenterology and Hepatology, VU University Medical Centre, Amsterdam 1081 HV, The Netherlands. [11]Department of Gastroenterology and Hepatology, University Medical Centre St. Radboud, Nijmegen 6525 GA, The Netherlands. [12]Department of Gastroenterology and Hepatology, Amsterdam Medical Centre, Amsterdam 1105 AZ, The Netherlands. [13]Department of Gastroenterology and Hepatology, University Medical Centre Utrecht, 3584 cXUtrecht, The Netherlands. [14]Department of Gastroenterology and Hepatology, University Medical Centre Maastricht, Maastricht 6229 HX, The Netherlands. [15]Department of Gastroenterology and Hepatology, Leiden University Medical Centre, Leiden 2333 ZA, The Netherlands. [16]Department of Gastroenterology and Hepatology, Erasmus Medical Centre, Rotterdam 3015 CE, The Netherlands. [17]Department of Gastroenterology and Hepatology, University of Groningen and University Medical Center Groningen, Hanzeplein 1, Groningen 9713 GZ, The Netherlands. [18]McGill University Centre for Molecular and Computational Genomics, 740 Dr. Penfield Avenue, Montreal H3A 0G1 QC, Canada. [19]UMR 1149 INSERM/Université Paris-Diderot Sorbonne Paris-Cité, Assistance Publique Hôpitaux de Paris, 48 Bd Sérurier, Paris 75019, France. [20]Department of Gastroenterology, University Hospital, De Pintelaan 185, Gent 9000, Belgium. [21]Translational Research in Gastrointestinal Disorders, Department of Clinical and Experimental Medicine, KU Leuven, UZ Herestraat 49, Leuven 3000, Belgium. [22]CHU-Liège and Unit of Gastroenterology, GIGA-R & Faculty of Medicine, University of Liège, 1 Avenue de l'Hôpital, Liège 4000, Belgium. These authors contributed equally: Yukihide Momozawa, Julia Dmitrieva. [†]A list of The International IBD Genetics Consortium members is provided below.

## The International IBD Genetics Consortium

Clara Abraham[23], Jean-Paul Achkar[24,25], Tariq Ahmad[26], Ashwin N. Ananthakrishnan[27,28], Vibeke Andersen[29,30,31], Carl A. Anderson[32], Jane M. Andrews[33], Vito Annese[34,35], Guy Aumais[36,37], Leonard Baidoo[38], Robert N. Baldassano[39], Peter A. Bampton[40], Murray Barclay[41], Jeffrey C. Barrett[32], Theodore M. Bayless[42], Johannes Bethge[43], Alain Bitton[44], Gabrielle Boucher[45], Stephan Brand[46], Berenice Brandt[43], Steven R. Brant[42], Carsten Büning[47], Angela Chew[48,49], Judy H. Cho[50], Isabelle Cleynen[21], Ariella Cohain[51], Anthony Croft[52], Mark J. Daly[53,54], Mauro D'Amato[55,56,57], Silvio Danese[58], Dirk De Jong[11], Goda Denapiene[59], Lee A. Denson[60], Kathy L. Devaney[27], Olivier Dewit[61], Renata D'Inca[62], Marla Dubinsky[63], Richard H. Duerr[38,64], Cathryn Edwards[65], David Ellinghaus[66], Jonah Essers[67,68], Lynnette R. Ferguson[69], Eleonora A. Festen[17], Philip Fleshner[70], Tim Florin[71], Andre Franke[66], Karin Fransen[72], Richard Gearry[41,73], Christian Gieger[74], Jürgen Glas[46,75], Philippe Goyette[45], Todd Green[54,67], Anne M. Griffiths[76], Stephen L. Guthery[77], Hakon Hakonarson[78], Jonas Halfvarson[78], Katherine Hanigan[52], Talin Haritunians[70], Ailsa Hart[79], Chris Hawkey[80], Nicholas K. Hayward[81], Matija Hedl[23], Paul Henderson[82,83], Xinli Hu[84], Hailiang Huang[53,54], Ken Y. Hui[50], Marcin Imielinski[39], Andrew Ippoliti[70], Laimas Jonaitis[85], Luke Jostins[86,87], Tom H. Karlsen[88,89,90], Nicholas A. Kennedy[91], Mohammed Azam Khan[92,93], Gediminas Kiudelis[85], Krupa Krishnaprasad[94], Subra Kugathasan[95], Limas Kupcinskas[96], Anna Latiano[34], Debby Laukens[20], Ian C. Lawrance[48,97], James C. Lee[98], Charlie W. Lees[91], Marcis Leja[99], Johan Van Limbergen[76], Paolo Lionetti[100], Jimmy Z. Liu[32], Gillian Mahy[101], John Mansfield[102], Dunecan Massey[98], Christopher G. Mathew[103,104], Dermot P.B. McGovern[70], Raquel Milgrom[105], Mitja Mitrovic[72,106], Grant W. Montgomery[81], Craig Mowat[107], William Newman[92,93], Aylwin Ng[27,108], Siew C. Ng[109], Sok Meng Evelyn Ng[23], Susanna Nikolaus[43], Kaida Ning[23], Markus Nöthen[110], Ioannis Oikonomou[23], Orazio Palmieri[34], Miles Parkes[98], Anne Phillips[107], Cyriel Y. Ponsioen[12], Uròs Potocnik[106,111], Natalie J. Prescott[103], Deborah D. Proctor[23], Graham Radford-Smith[52,112], Jean-Francois Rahier[113], Soumya Raychaudhuri[84], Miguel Regueiro[38], Florian Rieder[24], John D. Rioux[36,45], Stephan Ripke[53,54], Rebecca Roberts[41], Richard K. Russell[82], Jeremy D. Sanderson[114], Miquel Sans[115], Jack Satsangi[91], Eric E. Schadt[51], Stefan Schreiber[43,66], Dominik Schulte[43], L. Philip Schumm[116], Regan Scott[38], Mark Seielstad[117,118], Yashoda Sharma[23], Mark S. Silverberg[105], Lisa A. Simms[52], Jurgita Skieceviciene[85], Sarah L. Spain[32,119], A. Hillary Steinhart[105], Joanne M. Stempak[105], Laura Stronati[120], Jurgita Sventoraityte[94], Stephan R. Targan[70], Kirstin M. Taylor[114], Anje ter Velde[12], Leif Torkvist[121], Mark Tremelling[122], Suzanne van Sommeren[17], Eric Vasiliauskas[70], Hein W. Verspaget[15], Thomas Walters[76,123], Kai Wang[39], Ming-Hsi Wang[24,42], Zhi Wei[124], David Whiteman[81], Cisca Wijmenga[72], David C. Wilson[82,83], Juliane Winkelmann[125,126], Ramnik J. Xavier[27,54], Bin Zhang[51], Clarence K. Zhang[127], Hu Zhang[128,129], Wei Zhang[23], Hongyu Zhao[127] & Zhen Z. Zhao[81]

[23]Section of Digestive Diseases, Department of Internal Medicine, Yale School of Medicine, New Haven, CT, USA. [24]Department of Gastroenterology and Hepatology, Digestive Disease Institute, Cleveland Clinic, Cleveland, OH, USA. [25]Department of Pathobiology, Lerner Research Institute, Cleveland Clinic, Cleveland, OH, USA. [26]Peninsula College of Medicine and Dentistry, Exeter, UK. [27]Gastroenterology Unit, Massachusetts General Hospital, Harvard Medical School, Boston, MA 02114, USA. [28]Division of Medical Sciences, Harvard Medical School, Boston, MA, USA. [29]Focused Research Unit for Molecular Diagnostic and Clinical Research (MOK), IRS-Center Sonderjylland, Hospital of Southern Jutland, Åbenrå 6200, Denmark. [30]Institute of Molecular Medicine, University of Southern Denmark, Odense 5000, Denmark. [31]Institute of Regional Health Research, University of Southern Denmark, Odense, Denmark. [32]Wellcome Trust Sanger Institute, Wellcome Genome Campus, Hinxton, Cambridgeshire CB10 1SA, UK. [33]Inflammatory Bowel Disease Service, Department of Gastroenterology and Hepatology, Royal Adelaide Hospital, Adelaide, Australia. [34]Unit of Gastroenterology, Istituto di Ricovero e Cura a Carattere Scientifico-Casa Sollievo della Sofferenza (IRCCS-CSS) Hospital, San Giovanni Rotondo, Italy. [35]Strutture Organizzative Dipartimentali (SOD) Gastroenterologia 2, Azienda Ospedaliero Universitaria (AOU) Careggi, Florence, Italy. [36]Facult de Médecine, Universit de Montréal, Montréal, QC H3C 3J7, Canada. [37]Department of Gastroenterology, Hôpital Maisonneuve-Rosemont, Montréal, QC, Canada. [38]Division of Gastroenterology, Hepatology and Nutrition, Department of Medicine, University of Pittsburgh School of Medicine, Pittsburgh, PA 15213, USA. [39]Center for Applied Genomics, The Children's Hospital of Philadelphia, Philadelphia, PA, USA. [40]Department of Gastroenterology and Hepatology, Flinders Medical Centre and School of Medicine, Flinders University, Adelaide, Australia. [41]Department of Medicine, University of Otago, Christchurch, New Zealand. [42]Meyerhoff Inflammatory Bowel Disease Center, Department of Medicine, Johns Hopkins University School of Medicine, Baltimore, MD, USA. [43]Department for General Internal Medicine, Christian-Albrechts-University, Kiel, Germany. [44]Division of Gastroenterology, Royal Victoria Hospital, Montréal, QC, Canada. [45]Research Center, Montreal Heart Institute, Montréal, QC H1T 1C8, Canada. [46]Department of Medicine II, Ludwig-Maximilians-University Hospital Munich-Grosshadern, Munich, Germany. [47]Department of Gastroenterology, Campus Charité Mitte, Universitatsmedizin Berlin, Berlin, Germany. [48]Harry Perkins Institute for Medical Research, School of Medicine and Pharmacology, University of Western Australia, Murdoch, WA 6150, Australia. [49]IBD Unit, Fremantle Hospital, Fremantle, Australia. [50]Department of Genetics, Yale School of Medicine, New Haven, CT 06510, USA. [51]Department of Genetics and Genomic Sciences, Mount Sinai School of Medicine, New York, NY, USA. [52]Inflammatory Bowel Diseases, Genetics and Computational Biology, Queensland Institute of Medical Research, Brisbane, Australia. [53]Analytic and Translational Genetics Unit, Massachusetts General Hospital, Harvard Medical School, Boston, MA 02114, USA. [54]Broad Institute of MIT and Harvard, Cambridge, MA 02141, USA. [55]Clinical Epidemiology Unit, Department of Medicine Solna, Karolinska Institutet, Stockholm 17176, Sweden. [56]Department of Gastrointestinal and Liver Diseases, BioDonostia Health Research Institute, San Sebastián 20014, Spain. [57]IKERBASQUE, Basque Foundation for Science, Bilbao 48013, Spain. [58]IBD Center, Department of Gastroenterology, Istituto Clinico Humanitas, Milan, Italy. [59]Center of Hepatology, Gastroenterology and Dietetics, Vilnius University, Vilnius, Lithuania. [60]Pediatric Gastroenterology, Cincinnati Children's Hospital Medical Center, Cincinnati, OH, USA. [61]Department of Gastroenterology, Université Catholique de Louvain (UCL) Cliniques Universitaires Saint-Luc, Brussels, Belgium. [62]Division of Gastroenterology, University Hospital Padua, Padua, Italy. [63]Department of Pediatrics, Cedars Sinai Medical Center, Los Angeles, CA, USA. [64]Department of Human Genetics, University of Pittsburgh Graduate School of Public Health, Pittsburgh, PA 15261, USA. [65]Department of Gastroenterology, Torbay Hospital, Torbay, Devon, UK. [66]Institute of Clinical Molecular Biology, Christian-Albrechts-University of Kiel, Kiel 24118, Germany. [67]Center for Human Genetic Research, Massachusetts General Hospital, Harvard Medical School, Boston, MA, USA. [68]Pediatrics, Harvard Medical School, Boston, MA, USA. [69]Faculty of Medical & Health Sciences, School of Medical Sciences, The University of Auckland, Auckland, New Zealand. [70]F. Widjaja Foundation Inflammatory Bowel and Immunobiology Research Institute, Cedars-Sinai Medical Center, Los Angeles, CA 90048, USA. [71]Department of Gastroenterology, Mater Health Services, Brisbane, Australia. [72]Department of Genetics, University Medical Center Groningen, Groningen, The Netherlands. [73]Department of Gastroenterology, Christchurch Hospital, Christchurch, New Zealand. [74]Institute of Genetic Epidemiology, Helmholtz Zentrum München—German Research Center for Environmental Health, Neuherberg, Germany. [75]Department of Preventive Dentistry and Periodontology, Ludwig-Maximilians-University Hospital Munich-Grosshadern, Munich, Germany. [76]Division of Pediatric Gastroenterology, Hepatology and Nutrition, Hospital for Sick Children, Toronto, ON, Canada. [77]Department of Pediatrics, University of Utah School of Medicine, Salt Lake City, UT, USA. [78]Department of Gastroenterology, Faculty of Medicine and Health, Örebro University, SE-70182 Örebro, Sweden. [79]Department of Medicine, St. Mark's Hospital, Harrow, Middlesex, UK. [80]Nottingham Digestive Diseases Centre, Queens Medical Centre, Nottingham, UK. [81]Molecular Epidemiology, Genetics and Computational Biology, Queensland Institute of Medical Research, Brisbane, Australia. [82]Paediatric Gastroenterology and Nutrition, Royal Hospital for Sick Children, Edinburgh, UK. [83]Child Life and Health, University of Edinburgh, Edinburgh, Scotland, UK. [84]Division of Rheumatology Immunology and Allergy, Brigham and Women's Hospital, Boston, MA, USA. [85]Academy of Medicine, Lithuanian University of Health Sciences, Kaunas, Lithuania. [86]Wellcome Trust Centre for Human Genetics, University of Oxford, Headington OX3 7BN, UK. [87]Christ Church, University of Oxford, St Aldates OX1 1DP, UK. [88]Research Institute of Internal Medicine, Department of Transplantation Medicine, Division of Cancer, Surgery and Transplantation, Oslo University Hospital Rikshospitalet, Oslo, Norway. [89]Norwegian PSC Research Center, Department of Transplantation Medicine, Division of Cancer, Surgery and Transplantation, Oslo University Hospital Rikshospitalet, Oslo, Norway. [90]K.G. Jebsen Inflammation Research Centre, Institute of Clinical Medicine, University of Oslo, Oslo, Norway. [91]Gastrointestinal Unit, Western General Hospital University of Edinburgh, Edinburgh, UK. [92]Genetic Medicine, Manchester Academic Health Science Centre, Manchester, UK. [93]The Manchester Centre for Genomic Medicine, University of Manchester, Manchester, UK. [94]QIMR Berghofer Medical Research Institute, Royal Brisbane Hospital, Brisbane, Australia. [95]Department of Pediatrics, Emory University School of Medicine, Atlanta, GA, USA. [96]Department of Gastroenterology, Kaunas University of Medicine, Kaunas, Lithuania. [97]Centre for Inflammatory Bowel Diseases, Saint John of God Hospital, Subiaco, WA 6008, Australia. [98]Inflammatory Bowel Disease Research Group, Addenbrooke's Hospital, Cambridge CB2 0QQ, UK. [99]Faculty of medicine, University of Latvia, Riga, Latvia. [100]Dipartimento di Neuroscienze, Psicologia, Area del Farmaco e Salute del Bambino, Universitê di Firenze Strutture Organizzative Dipartimentali (SOD) Gastroenterologia e Nutrizione Ospedale Pediatrico Meyer, Firenze, Italy. [101]Department of Gastroenterology, The Townsville Hospital, Townsville, Australia. [102]Institute of Human Genetics, Newcastle University, Newcastle upon Tyne, UK. [103]Department of Medical and Molecular Genetics, King's College London, London SE1 9RT, UK. [104]Sydney Brenner Institute for Molecular Bioscience, University of the Witwatersrand, Johannesburg 2193, South Africa. [105]Inflammatory Bowel Disease Centre, Mount Sinai Hospital, Toronto, ON, Canada. [106]Center for Human Molecular Genetics and Pharmacogenomics, Faculty of Medicine, University of Maribor, Maribor, Slovenia. [107]Department of Medicine, Ninewells Hospital and Medical School, Dundee, UK. [108]Center for Computational and Integrative Biology, Massachusetts General Hospital, Harvard Medical School, Boston, MA, USA. [109]Department of Medicine and Therapeutics, Institute of Digestive Disease, Chinese University of Hong Kong, Hong Kong, Hong Kong. [110]Department of Genomics Life & Brain Center, University Hospital Bonn, Bonn, Germany. [111]Faculty for Chemistry and Chemical Engineering, University of Maribor, Maribor, Slovenia. [112]Department of Gastroenterology, Royal Brisbane and Womens Hospital, Brisbane, Australia. [113]Department of Gastroenterology, Université Catholique de Louvain (UCL) Centre Hospitalier Universitaire (CHU) Mont-Godinne, Mont-Godinne, Belgium. [114]Department of Gastroenterology, Guy's & St. Thomas' NHS Foundation Trust, St.-Thomas Hospital, London, UK. [115]Department of Digestive Diseases, Hospital Quiron Teknon, Barcelona, Spain. [116]Department of Public Health Sciences, University of Chicago, Chicago, IL, USA.

[117]Human Genetics, Genome Institute of Singapore, Singapore, Singapore. [118]Institute for Human Genetics, University of California, San Francisco, CA, USA. [119]Open Targets, Wellcome Trust Genome Campus, Hinxton, Cambridgeshire CB10 1SD, UK. [120]Department of Biology of Radiations and Human Health, Agenzia Nazionale per le Nuove Tecnologie l'energia e lo Sviluppo Economico Sostenibile (ENEA), Rome, Italy. [121]Department of Clinical Science Intervention and Technology, Karolinska Institutet, Stockholm, Sweden. [122]Gastroenterology & General Medicine, Norfolk and Norwich University Hospital, Norwich, UK. [123]Faculty of Medicine, University of Toronto, Toronto, ON, Canada. [124]Department of Computer Science, New Jersey Institute of Technology, Newark, NJ, USA. [125]Institute of Human Genetics, Technische Universität München, Munich, Germany. [126]Department of Neurology, Technische Universität München, Munich, Germany. [127]Department of Biostatistics, School of Public Health, Yale University, New Haven, CT, USA. [128]Department of Gastroenterology, West China Hospital, Chengdu, Sichuan, China. [129]State Key Laboratory of Biotherapy, Sichuan University West China University of Medical Sciences (WCUMS), Chengdu, Sichuan, China.

