## [Peer Review File · Nature Communications]

Reviewer #1 (Remarks to the Author):

In their manuscript, Momozawa and colleagues generated, in addition to genome-wide Illumina SNP array data, Illumina HT12 array-based transcriptome datasets of nine different cell types from peripheral blood and intestinal biopsies for 323 healthy Europeans (CEDAR dataset) and identified a total of 23,650 cis-eQTLs (FDR <0.05).

First, from the CEDAR dataset, they generated Expression Association Patterns (EAP; i.e. the $-\log(p)$ values of association for the SNPs surrounding one or more genes). Based on similarity measures between EAPs in different regions and tissues they clustered 9,720 distinct “cis-regulatory modules” (cRM) including at least 1 gene and tissue. 68% of cRMs were gene- and tissue-specific, 22% were gene-specific but operating across multiple tissues, 10% were multi-genic and nearly always multi-tissular.

Second, they determined strong correlations between EAP from the CEDAR dataset and “disease association patterns” (DAP) obtained from the International IBD Genetics Consortium (IIBDGC) SNP array data. Strong correlations between DAP and EAP (similarity ≥ 0.6) were observed for at least 63 IBD risk loci, involving 99 genes, with regulatory variants preferentially driving multigenic modules. There was no enrichment of specific cell-types amongst correlated EAP ($p \geq 0.11$).

Third, they performed targeted sequencing for 555 coding exons of 38 genes selected amongst those with strongest DAP-EAP correlations, plus 7 genes with suggestive DAP-EAP evidence backed by literature, in 6,597 European CD cases and 5,502 matched controls. 18 of these genes were part of single-gene cRM, the remaining 27 genes corresponded to multi-gene cRM mapping to 15 IBD risk loci. They identified a total of 174 loss-of-function (LoF) variants, 2,567 missense variants and performed gene-based association burden tests for rare variants (MAF <0.005) from loci where Cis-eQTLs match known IBD association signals.

I am missing a short paragraph at the beginning of the manuscript explaining the main idea of the introduced methods and studies. In my view, the overall intention of the performed studies is hard to understand.

I have doubts that this study acquired new findings by combining the CEDAR dataset with genotype data published by previous IIBDGC studies. What is new in Table 1 in comparison to eQTL results which were published in previous large-scale IIBDGC studies? Which datasets from the IIBDGC were used to calculate DAPs? In 2012, Jostins and colleagues determined that “64 IBD-associated SNPs are in linkage disequilibrium with variants known to regulate gene expression” (Jostins et al. 2012 PubMedID 23128233). Jostins et al. combined GWAS data with eQTL data from more than 1000 individuals, including data from both immune and non-immune tissue datasets. What is the overlap and what is the difference between the 64 GWAS risk loci previously identified in Jostins et al. and the 63 IBD risk loci that were identified in this study? Also recently, further 12 eQTL datasets were

searched to identify variants within IBD risk loci that are associated with variation in gene expression (De Lange et al. 2016 PubMedID 28067908). What is new when comparing results to De Lange et al.?

Why do the authors think that rare LoF and missense variants are responsible for a large fraction of regulatory eQTLs effects within established IBD GWAS risk loci? As stated in the Abstract, previous work showed that “corresponding causative genes have been identified for only a minority of loci (approx. 10) on the basis of independently associated coding variants” (Huang et al. 2017 Nature, PubMed 28658209) and the authors speak about the “regulatory nature of the majority of common risk variants”.

Momozawa and colleagues identified that for NOD2 ($p = 6.9 \times 10^{-7}$) and IL23R ($p = 1.8 \times 10^{-4}$) LoF and damaging variants were significantly enriched in respectively cases and controls, as previously shown by finemapping of IBD risk loci (Huang et al. 2017 Nature, PubMed 28658209). However, despite a significant shift towards lower p-values when considering all resequenced 45 genes jointly, none of these were individually significant when accounting for multiple testing (Figure 3b). The authors further explored three modifications to increase the power of their burden test and designed a module- rather than gene-based burden test. However, neither of the three approaches could improve the results. Finally, the authors estimated that ≥ 10 -fold larger sample sizes will be needed in the future to achieve adequate power if using the same approach.

Minor comments

- Are values 22% (blue bars) and 10% (red bars) accidentally mixed-up in the main text and the legend of Figure 1b? In Figure 1b it looks that the proportion of blue bars is much smaller compared to the fraction represented by the red bars.
- Figure 1c: What is the difference between "->" arrows (“with detectable expression but without evidence for cis-eQTL”) and a non-significant "small arrow". Why making this distinction? In both cases, there is no statistical evidence of observing a cis-eQTL. I am further missing an explanation for the green arrow. Is the result really informative and significant for the reader if only one single significant “down arrow” is shown as for LRP11? How can you conclude that the LATS1 gene is not affected by the same regulatory variants if no expression at all could be detected for genes marked by "-" ?
- Suppl Fig 1 description does not fit to the sentence from main manuscript where it's referenced.

Reviewer #2 (Remarks to the Author):

This is an elegant study generating a large transcriptome dataset from nine cell types in which 23,650 cis-eQTL are identified. SNP “expression association patterns” (EAP) are defined to associate these cis eQT’s with ~9,720 modules of regulatory variants. The authors fail to show statistical significance of an association between variants in re-sequenced CD candidate genes and eQTL due to power restrictions caused by multiple testing and limitations of sample size in their specific setting.

Major points:

1. Given the fact that regulatory variants are in the focus of interest why has an exome based re-sequencing approach been chosen? The starting hypothesis of the study was that regulatory variants are probably more relevant than the two handful of missense variants identified in IBD but at the end the focus of the paper is on missense variants.
2. What was the motivation for the choice of tissues used for transcriptome analysis. For example, why thrombocytes and why not mucosal epithelial cells if biopsies were available? Why were no patient samples with known genotypes used?
3. The main problem is that the interesting hypothesis raised is not confirmed. This raises the question of conceptual novelty in comparison to previous eQTL studies (Jostins et al, 2012). Clearly the biomaterials used here are superior but do not result in novel significant findings. At least a thorough discussion and comparison with the eQtl results reported by Jostins and colleagues as well as De Lange et al. 2016 should be provided.
4. The lack of individual significance in the resequencing experiment in 45 susceptibility genes raises important questions about the burden concept brought forward by the authors. The estimation of a 10 x larger sample size is a good guidance for the next experiment that however needs to be done.

Reviewer #3 (Remarks to the Author):

This paper describes eQTL analysis of multiple tissues (immune and bowel biopsies) to define modules of co-regulated genes and their shared regulatory variants, before integration with IBD GWAS data in order to define IBD candidate genes. 45 of these genes were resequenced in ~12000 samples, which showed enrichment for disease association under burden tests. This is a nice idea -

to use eQTLs to identify disease genes and then confirm that these genes (rather than proxies) are causal in disease through sequencing. The authors note, however, that much larger sample sizes than currently available would be needed to confirm disease association through rare variant tests.

MAJOR CONCERNS

The authors need to clearly state how this analysis differs from 'Huang,H. et al. (2017) Fine-mapping inflammatory bowel disease loci to single-variant resolution. Nature'. It seems to include the same eQTL datasources and same senior authors. How do the authors justify the very different conclusions between this manuscript and Huang et al. findings (given that there are shared authors between the two papers). Huang et al. find only a modest overlap between eQTL and fine mapped UC and CD variants, in comparison with this paper which finds overlap in 63 IBD disease risk loci

The structure of the paper is, in my opinion, compromised and the supplementary data abused. This makes the paper quite hard to follow in it's current format. It appears that the supplementary material could, on its own, form a paper describing the eQTL analysis and definition of cRMs. As so much of the results are based on details in the supplementary, some better summaries in the main text is needed, eg on conceptually what theta captures.

I am not clear where the value of this paper lies - in the eQTL resource and cRMs or in the candidacy of genes for Crohn's disease thus identified? If the former, then it is disappointing the methods and results are presented so briefly, that the modules are not characterised in any detail, and that the complete results (not just eQTLs with $FDR < 0.05$) are not simply made available for download rather than obscured through the interactive browser. If the latter, I'm not convinced this is so novel, as many of the genes listed are already established causal genes - indeed, that is stated as support for their list. What new knowledge of IBD has been gained? Myself, I believe the cRMs are of potentially more widespread utility if robustly defined (more on this below), but greater detail on their definition, and evidence for their robustness, would be needed for such value to be realised.

The authors introduce the 'theta' method for examining the overlap between EAP's. This is a critical part of the manuscript however it is given only a cursory exposition in Figure 1A. The method is defined in some detail in the supplementary, but no justification for constants used is supplied. How were they chosen?

There is no evidence that the method produces valid results. How is the metric affected by LD? Intuitively I would expect regions with lots of LD to have more chance to have a high 'theta' as the regulatory variant even if different between EAP's would be more likely to be correlated. There is no comparison with existing colocalization methods (eg those used in Huang et al) on simulated data. No details are given on how cRM were defined to be tissue specific or not in main text. Finally, all reported numbers completely reflect decisions made to choose parameter values, none of which

are justified - eg $\theta > 0.6$, exclusion of SNPs with $-\log(p) < 1.3$, $k=30$, $T=0.3$, $P=1$. What happens to the conclusions of the manuscript if any/all of these are varied?

The notion (line 99) that snps are enriched in multi-genic modules again could result from confounding by LD, as surely multi-genic CRM correspond to longer regions of LD. I don't think that the approach detailed in section 7 of supplementary material adequately addresses this.

Data availability: There is a web portal but where are they submitting the raw genotypes and transcriptomics data so the whole scientific community can benefit from this impressive resource?

MINOR CONSIDERATIONS

eQTLs are from 323 samples across 9 cell types. Average was 56, with a range 19-86, across which the immune system is known to vary. How were age effects considered? Also ascertainment bias - the biopsies were a screening process but I assume the cohort must have had some preexisting reason to undergo such invasive surgery. Information about this needs to be provided.

Why use SIFT and Pphen2 what about an ensemble method such as CADD ?

Line 80 - how are the genes 'closely linked' - do the authors mean within the same linkage disequilibrium block ? If so how do they define an LD block ?

Line 88 - multi-tissular please change to multi-tissue.

Line 98 - Can you compute a statistic for whether this sharing between cell types is significant. Supp Fig 2B would be much easier to interpret as a heatmap or as a hierarchical cluster plot (which would also be a nicer demonstration of reconstituting ontogenic relations).

Line 112 - The website is a useful tool however I felt it could do with a more complete help section. The Karyotype browser is difficult to interpret as there are no labels that have biological meaning outside the arbitrary CRM identifiers - perhaps convert to karyotype bands (as human geneticists will have more chance to interpret these), I discourage the use of gene names as this can mislead as to which is the causal gene.

Line 123 - I think that the statement that the burden of rare disruptive variants is expected to differ between cases and controls needs a citation to back it up. For example, in T2D Fuchsberger, C. et al. (2016) The genetic architecture of type 2 diabetes. Nature.

Line 130 - What is the allelic spectrum of the variants found ? How do these compare with GNOMAD.

Line 152 - What is superscript f.i.3 ?

Line 203 - I think the evidence for these conclusions is weak also the authors should say which statistical test was performed.

Reviewers' comments:

REVIEWER #1 (REMARKS TO THE AUTHOR):

In their manuscript, Momozawa and colleagues generated, in addition to genome-wide Illumina SNP array data, Illumina HT12 array-based transcriptome datasets of nine different cell types from peripheral blood and intestinal biopsies for 323 healthy Europeans (CEDAR dataset) and identified a total of 23,650 cis-eQTLs (FDR <0.05). First, from the CEDAR dataset, they generated Expression Association Patterns (EAP; i.e. the $-\log(p)$ values of association for the SNPs surrounding one or more genes). Based on similarity measures between EAPs in different regions and tissues they clustered 9,720 distinct “cis-regulatory modules” (cRM) including at least 1 gene and tissue. 68% of cRMs were gene- and tissue-specific, 22% were gene-specific but operating across multiple tissues, 10% were multi-genic and nearly always multi-tissular. Second, they determined strong correlations between EAP from the CEDAR dataset and “disease association patterns” (DAP) obtained from the International IBD Genetics Consortium (IIBDGC) SNP array data. Strong correlations between DAP and EAP (similarity ≥ 0.6) were observed for at least 63 IBD risk loci, involving 99 genes, with regulatory variants preferentially driving multigenic modules. There was no enrichment of specific cell-types amongst correlated EAP ($p \geq 0.11$).

Third, they performed targeted sequencing for 555 coding exons of 38 genes selected amongst those with strongest DAP-EAP correlations, plus 7 genes with suggestive DAP-EAP evidence backed by literature, in 6,597 European CD cases and 5,502 matched controls. 18 of these genes were part of single-gene cRM, the remaining 27 genes corresponded to multi-gene cRM mapping to 15 IBD risk loci. They identified a total of 174 loss-of-function (LoF) variants, 2,567 missense variants and performed gene-based association burden tests for rare variants (MAF < 0.005) from loci where Cis-eQTLs match known IBD association signals.

I am missing a short paragraph at the beginning of the manuscript explaining the main idea of the introduced methods and studies. In my view, the overall intention of the performed studies is hard to understand.

We thank the reviewer for this comment. It stems in part from the fact that our manuscript was written in Nature Genetics Letter format (with strict restrictions on allowed characters/words), and forwarded “as is” to Nature Communications. We have now added an introductory section that should help the reader better understand the background of our study and its rationale (lines 67 - 149).

I have doubts that this study acquired new findings by combining the CEDAR dataset with genotype data published by previous IIBDGC studies. What is new in Table 1 in comparison to eQTL results which were published in previous large-scale IIBDGC studies? Which datasets from the IIBDGC were used to calculate DAPs? In 2012, Jostins and colleagues determined that “64 IBD-associated SNPs are in linkage disequilibrium with variants known to regulate gene expression” (Jostins et al. 2012 PubMedID 23128233). Jostins et al. combined GWAS data with eQTL data from more than 1000 individuals, including data from both immune and non-immune tissue datasets. What is the overlap and what is the difference between the 64 GWAS risk loci previously

identified in Jostins et al. and the 63 IBD risk loci that were identified in this study? Also, recently, further 12 eQTL datasets were searched to identify variants within IBD risk loci that are associated with variation in gene expression (De Lange et al. 2016 PubMedID 28067908). What is new when comparing results to De Lange et al.?

We have added a column to Table 1 that refers to the publications by others, reporting coincident disease-eQTL associations for the same gene(s). We can see from this that coincident DAP-EAP were not previously reported for 47 of the 99 reported positional candidates, hence demonstrating that we really contribute novel finding in this study. In addition, we have added a supplemental table (Suppl. Table 5) in which we report coincident disease-eQTL associations reported by others that we do not included in our Table 1. We provide information about the reasons why we excluded them from our final list. It shows that several of those are likely genuine but just missed our thresholds. We have modified the manuscript accordingly (lines 245-255).

Why do the authors think that rare LoF and missense variants are responsible for a large fraction of regulatory eQTLs effects within established IBD GWAS risk loci? As stated in the Abstract, previous work showed that “corresponding causative genes have been identified for only a minority of loci (approx. 10) on the basis of independently associated coding variants” (Huang et al. 2017 Nature, PubMed 28658209) and the authors speak about the “regulatory nature of the majority of common risk variants”.

We do not believe that rare LoF and missense variants are responsible for a large fraction of regulatory effects within established IBD GWAS risk loci. The majority of causative variants driving GWAS signals are known to be common regulatory variants. Thus, these have to cause cis-eQTL effects for one (or more) gene(s) in one (or more) disease relevant cell types. We use our eQTL dataset and the “colocalisation” method that we have developed to identify matching “disease association patterns” and “eQTL association patterns”. This allowed us to identify 99 strong, eQTL-supported, positional candidate genes in 63 out of 200 studied GWAS-identified risk loci. In a second step, we attempt to use “burden tests”, i.e. tests that search for a differential burden of rare and low frequency disruptive coding variants between cases and controls, in 45 of the 100 candidate genes in order to prove their causality. The common (regulatory) variants that underlie the GWAS and eQTL signals are thus different from the rare/low frequency variants that are used in the burden test (and that is what we want, as we want both signals to be independent and hence to strengthen each other). The introductory section mentioned above, should now have clarified our strategy, and hopefully be of interest for a broad readership (lines 67 - 149).

Momozawa and colleagues identified that for NOD2 ($p = 6.9 \times 10^{-7}$) and IL23R ($p = 1.8 \times 10^{-4}$) LoF and damaging variants were significantly enriched in respectively cases and controls, as previously shown by fine-mapping of IBD risk loci (Huang et al. 2017 Nature, PubMed 28658209).

It is noteworthy that the NOD2/IL23R signal that we identify in this study is distinct from that in Huang et al. The coding variants identified in Huang et al. are

common or low frequency (MAF ≥ 0.005), while the ones driving the signal of the burden test in this study are other variants and all rare (MAF ≤ 0.005).

However, despite a significant shift towards lower p-values when considering all resequenced 45 genes jointly, none of these were individually significant when accounting for multiple testing (Figure 3b). The authors further explored three modifications to increase the power of their burden test and designed a module- rather than gene-based burden test. However, neither of the three approaches could improve the results. Finally, the authors estimated that ≥ 10 -fold larger sample sizes will be needed in the future to achieve adequate power if using the same approach.

Minor comments

- Are values 22% (blue bars) and 10% (red bars) accidentally mixed-up in the main text and the legend of Figure 1b? In Figure 1b it looks that the proportion of blue bars is much smaller compared to the fraction represented by the red bars.

There was no accidental mix-up. The legend stated that “*The number of observations for single-gene cRM were divided by 10 in the graph for clarity.*”, and this was also indicated in the graph: “x10”. We have added “*Thus, there are more cases of single-gene, multi-tissue cRM (blue; 2,155) than multi-gene cRM (red; 967).*” to the legend for clarity (line 629).

- Figure 1c: What is the difference between “->” arrows (“with detectable expression but without evidence for cis-eQTL”) and a non-significant “small arrow”. Why making this distinction? In both cases, there is no statistical evidence of observing a cis-eQTL. I am further missing an explanation for the green arrow. Is the result really informative and significant for the reader if only one single significant “down arrow” is shown as for LRP11? How can you conclude that the LATS1 gene is not affected by the same regulatory variants if no expression at all could be detected for genes marked by “-” ?

We have modified Figure 1C to, hopefully, make it clearer. Specifically, we have made a link between the coloured arrows in the main part of the figure (corresponding to cRM 57 (yellow) and cRM 3694 (green)) and the “inset”. The red and blue colours are now only used in the inset, for low and high values of ϑ , respectively. With regards to the distinction between “->” and “small arrows”. It is correct that both cases are characterized by eQTL FDR values > 0.05 . This either means that there is no eQTL (true negative), or that there is an eQTL but that the q-value does not exceed the chosen significance threshold. In the first instance (true negative), there is no reason to expect high $|\vartheta|$ values with any other EAP. However, if we observe such high $|\vartheta|$ values, it strongly suggests that we are dealing with a real eQTL (false negative). This is the basis of the distinction that we are making: “->” in the first instance, “small arrow” in the second instance. We have modified the legend accordingly (line 632).

With regards to the two green arrows ... By bordering the ϑ values in the inset in yellow (cRM 57) and green (cRM 3694), respectively, as well as adding a green arrow to highlight cRM 3694 in the inset, we hope to make the figure more easily interpretable.

We don't find evidence for expression of the LATS1 gene in any of the nine analysed cell types. Hence, and by definition, the LATS1 gene cannot be considered to be regulated by any variant in these cell types (there is no difference in LATS1 expression level between SNP genotypes – expression levels are “zero” for all genotypes). It does seem reasonable for us to distinguish this situation (“-”) from the situation with expression but no evidence for an eQTL (“->”).

- Suppl Fig 1 description does not fit to the sentence from main manuscript where it's referenced.

We have inserted a paragraph to better explain ϑ and the clustering approach following the recommendation of reviewer 3. Suppl. Fig 1 (now Suppl. Fig. 2) is now referred to line 190 and line 196, which should be more appropriate.

REVIEWER #2 (REMARKS TO THE AUTHOR):

This is an elegant study generating a large transcriptome dataset from nine cell types in which 23,650 cis-eQTL are identified. SNP “expression association patterns” (EAP) are defined to associate these cis eQT's with ~9,720 modules of regulatory variants. The authors fail to show statistical significance of an association between variants in re-sequenced CD candidate genes and eQTL due to power restrictions caused by multiple testing and limitations of sample size in their specific setting.

We thank the reviewer for his appreciation of our study.

Major points:

1. Given the fact that regulatory variants are in the focus of interest why has an exome based re-sequencing approach been chosen? The starting hypothesis of the study was that regulatory variants are probably more relevant than the two handful of missense variants identified in IBD but at the end the focus of the paper is on missense variants.

This comment is related to the first comment of reviewer 1. Being given more space than in Nature Genetics (from which this manuscript was forwarded), we have now added an introductory section that explains the motivation and rationale of the chosen strategy and why investigate both common regulatory variants and rare missense variants. We hope that this will respond in a satisfactory way to the present question (lines 67 - 149).

2. What was the motivation for the choice of tissues used for transcriptome analysis. For example, why thrombocytes and why not mucosal epithelial cells if biopsies were available? Why were no patient samples with known genotypes used?

The motivation was to a large extent practical. It was possible, and approved by the ethical committee, to sample blood in sufficient quantities to be able to fractionate and obtain sufficient cell numbers for six blood cell populations. At least five of these are known to play key roles in innate and acquired immunity

and are therefore - in our opinion - potentially relevant when studying the pathogenesis of IBD. The fact that we also obtained intestinal biopsies at three anatomical locations is quite exceptional. We are now clarifying this in the manuscript (lines 158-162).

It would indeed have been very interesting to separate cell populations from the biopsies (such as epithelial cells as suggested by the reviewer), but - because of major technical and logistic challenges associated with it - we have not been able to do this as part of the present study. It is noteworthy that we are planning to repeat a similar study using single cell sequencing, but this is completely beyond the scope of the present study.

The reason why we use healthy individuals rather than patients is introduced in the introduction (lines 107-113). We have also added a paragraph in the discussion to address this question (lines 397-408).

3. The main problem is that the interesting hypothesis raised is not confirmed. This raises the question of conceptual novelty in comparison to previous eQTL studies (Jostins et al, 2012). Clearly the biomaterials used here are superior but do not result in novel significant findings. At least a thorough discussion and comparison with the eQTL results reported by Jostins and colleagues as well as De Lange et al. 2016 should be provided.

This comment is also related to a question raised by reviewer 1. We have added a column to Table 1 that refers to the publications by others, reporting coincident disease-eQTL associations for the same gene(s). We can see from this that coincident DAP-EAP were not previously reported for 47 of the 99 reported positional candidates, hence demonstrating that we really contribute novel finding in this study. In addition, we have added a supplemental table (Suppl. Table 5) in which we report coincident disease-eQTL associations reported by others that we do not included in our Table 1. We provide information about the reasons why we excluded them from our final list. It shows that several of those are likely genuine but just missed our thresholds. We have modified the manuscript accordingly (lines 245-255).

We thank the reviewer for emphasizing the value of this new dataset.

4. The lack of individual significance in the resequencing experiment in 45 susceptibility genes raises important questions about the burden concept brought forward by the authors. The estimation of a 10 x larger sample size is a good guidance for the next experiment that however needs to be done.

We agree with this comment. We hope in this regard that the last section of the discussion (lines 435-450) will be of interest to the readers.

REVIEWER #3 (REMARKS TO THE AUTHOR):

This paper describes eQTL analysis of multiple tissues (immune and bowel biopsies) to define modules of co-regulated genes and their shared regulatory variants, before integration with IBD GWAS data in order to define IBD candidate genes. 45 of these genes were resequenced in ~12000 samples, which showed enrichment for disease

association under burden tests. This is a nice idea - to use eQTLs to identify disease genes and then confirm that these genes (rather than proxies) are causal in disease through sequencing. The authors note, however, that much larger sample sizes than currently available would be needed to confirm disease association through rare variant tests.

We thank the reviewer for his supportive comments. We note that reviewer approves of the selected strategy.

MAJOR CONCERNS

The authors need to clearly state how this analysis differs from 'Huang,H. et al. (2017) Fine-mapping inflammatory bowel disease loci to single-variant resolution. Nature'. It seems to include the same eQTL datasources and same senior authors. How do the authors justify the very different conclusions between this manuscript and Huang et al. findings (given that there are shared authors between the two papers). Huang et al. find only a modest overlap between eQTL and fine mapped UC and CD variants, in comparison with this paper which finds overlap in 63 IBD disease risk loci.

There are two main reasons why the numbers differ between this study and Huang et al. (2017). The first is that in this study we analysed 200 IBD risk loci, while in Huang et al. we analysed only 97. The second is that we corrected the gene expression data with principal components (PC) to correct for hidden confounders in the expression data. Following Fairfax et al. (2014), we included the number of PC in the model that maximized the number of significant eQTL for each tissue. By doing so we increased the number of significant eQTL by ~2 (11,964 to 23,650). As an example, we analysed 480 eQTL in the 97 IBD risk loci in Huang et al. (2017) (see Methods - Processing and quality control of new eQTL ULg dataset), while this number was increased to 880 usable eQTL in this study.

Using the earlier version of the CEDAR dataset (no correction with PC, 480 eQTL), Huang et al. reported significant (albeit modest) enrichment of overlapping disease and eQTL signals for CD4, ileum, colon and rectum, focusing on 76 of 97 studied IBD risk loci (MAF of disease variant > 0.05) (Extended table 2 in Huang et al.). Using the new version of the CEDAR dataset (880 eQTL), we repeated the enrichment analysis focusing on 63 (the fact that this number is the same as in Table 1 is fortuitous) of the same 97 IBD loci (CD risk loci; MAF of disease variant > 0.05). We obtained strong evidence for enrichment in CD4 and CD8. We are now reporting this analysis in the main text (lines 2566-267), in Suppl. Methods (lines 316-379) and in Suppl. Table 6.

The structure of the paper is, in my opinion, compromised and the supplementary data abused. This makes the paper quite hard to follow in its current format. It appears that the supplementary material could, on its own, form a paper describing the eQTL analysis and definition of cRMs. As so much of the results are based on details in the supplementary, some better summaries in the main text is needed, eg on conceptually what theta captures.

We agree in part with this comment. As stated in our response to the other reviewers, this is due to the fact that this manuscript was directly forwarded to Nature Communications from Nature Genetics which imposes strict constraints on allowed characters/words. To make the manuscript easier to read we have now added an introduction explaining the motivation and rationale of the paper, and are providing more information about the methods in the main text (in addition to the supplementary material). As an example, a whole paragraph is now devoted to briefly explain ϑ and the clustering method (lines 172-190). A more detailed description is still made available in Suppl. Methods. The eQTL analysis per se is rather standard. We only very briefly refer to the method used in the main text (lines 160-162). We still think that it is important that the reader can find a very detailed description of what has been done exactly in Suppl. Methods.

I am not clear where the value of this paper lies - in the eQTL resource and cRMs or in the candidacy of genes for Crohn's disease thus identified? If the former, then it is disappointing the methods and results are presented so briefly, that the modules are not characterised in any detail, and that the complete results (not just eQTLs with FDR<0.05) are not simply made available for download rather than obscured through the interactive browser. If the latter, I'm not convinced this is so novel, as many of the genes listed are already established causal genes - indeed, that is stated as support for their list. What new knowledge of IBD has been gained? Myself, I believe the cRMs are of potentially more widespread utility if robustly defined (more on this below), but greater detail on their definition, and evidence for their robustness, would be needed for such value to be realised.

We can understand the comment made by the reviewer - it is a question of publication strategy. We prefer to produce publications with a lot of content rather than fragment the work in multiple manuscripts, but we accept that this approach be questioned. We think that the value of this publication indeed lies in at least four main contributions: (i) a new very large eQTL dataset (all the data will be made available without any restrictions upon acceptance of the publication - we mentioned this upon submission, it is now stated explicitly in the manuscript (lines 489-491), (ii) the notion of cRM and the identification of 9,720 of these, using a new approach (based on ϑ) to identify and cluster EAP driven by the same regulatory variants, (iii) the identification of 99 eQTL-supported positional candidate genes for IBD of which 47 are new, and (iv) the demonstration that the burden test as a way to formally prove gene causality for common complex diseases has issues even when applied to cohorts as large as 12,000 individuals.

We disagree with the statement of the reviewer that "*many of the genes listed are already established causal genes*". This is in our opinion not correct. They are strong candidates (this is the whole point of studies like this one), but their causality is not proven (see also our Introduction).

We also disagree with the reviewer that we didn't present evidence for the robustness of our method. We assume that this is due to the fact that this was all described in Supplemental Material only. There is an entire section (Suppl. Material, lines 225-276) and specific Suppl. Fig. (now Suppl. Fig. 1) devoted to

evaluating the distribution of ϑ under H_0 and H_1 . This was used to select the threshold value of $|\vartheta| \geq 0.6$, and to obtain the corresponding false positive and false negative rates. We are now referring to this analysis in the main manuscript as well (lines 181-186).

The authors introduce the ‘theta’ method for examining the overlap between EAP’s. This is a critical part of the manuscript however it is given only a cursory exposition in Figure 1A.

As mentioned above, we have now added a paragraph describing the method in the main section of the manuscript (lines 172-190).

The method is defined in some detail in the supplementary, but no justification for constants used is supplied. How were they chosen?

See comment above with regards to the choice of 0.6 as threshold value for $|\vartheta|$. See below for other constants.

There is no evidence that the method produces valid results. How is the metric affected by LD? Intuitively I would expect regions with lots of LD to have more chance to have a high ‘theta’ as the regulatory variant even if different between EAP’s would be more likely to be correlated.

The metric is not affected by LD, but the probability to obtain a given value of $|\vartheta|$ by chance alone is indeed affected by the extend of LD in the region. This was addressed by generating locus-specific distributions of ϑ under the null hypothesis, by simulating eQTL explaining the same variance as the true QTL but driven by all possible variants in the region. This allows one to compute locus- and eQTL-specific empirical p-values to obtain the corresponding $|\vartheta|$ value by chance alone, accounting for the local LD structure. This was described in Supplemental Methods lines 308-315 and 230-269 (generating in silico eQTL). The corresponding empirical p-values for DAP-EAP with $|\vartheta| \geq 0.6$ are now given in Table 1 and Suppl. Table 4.

To enhance clarity of the manuscript, we are now referring to this in the main section of the manuscript (lines 224-233).

There is no comparison with existing colocalization methods (eg those used in Huang et al) on simulated data.

To provide a comparison with other methods on exactly the same dataset, we have reanalysed our data using the SMR approach (Zhu et al., 2016). Suppl. Tables 4 now provides information about SMR results for the genes that were detected by our ϑ -based approach, while Suppl. Table 5 now also reports genes that were found by SMR (applied to our data) but not by the ϑ -based approach. The corresponding results are reported in the main text (lines 245-255).

No details are given on how cRM were defined to be tissue specific or not in main text.

We have added two sentences in the main text to clarify this (lines 196-199).

Finally, all reported numbers completely reflect decisions made to choose parameter values, none of which are justified - eg $\theta > 0.6$, exclusion of SNPs with $-\log(p) < 1.3$, $k=30$, $T=0.3$, $P=1$. What happens to the conclusions of the manuscript if any/all of these are varied?

As stated above, the choice of 0.6 as threshold for $|\vartheta|$ was extensively justified based on the known distributions of $|\vartheta|$ under H_0 and H_1 . It was accompanied with a rigorous estimate of false positive and false negative rate. We understand that the reviewer missed this as this analysis was not mentioned in the main section of the manuscript, only in Suppl. Methods. We now refer to these analyses in the main text, lines 181-186. This analysis quantifies the robustness of our method in a standard and rigorous manner.

Variants with $-\log(p) < 1.3$ for both EAP (or for the EAP and the DAP) were ignored because the majority of them are by definition not associated with either trait. There is therefore no reason to expect that they could contribute useful information to the correlation metric: their ranking in terms of $-\log(p)$ values becomes more and more random as the $-\log(p)$ decreases. We have now added this justification to Suppl. Methods (lines 184-189).

K and T. The point here is that if two association patterns are “similar” (driven by the same variants), the correlation (r_w in Suppl. Methods) between $-\log(p)$ values is expected to be positive. If two association patterns are different (driven by distinct variants) they may generate strong negative correlations (r_w). The first part of the method aims at weeding out such instances (negative r_w). One way to do this is to choose a simple threshold value for r_w . We herein propose an approach that offers more flexibility: it generates a penalty that increases when the correlation decreases with an adaptable rate. As shown in Suppl. Fig. 8, the values of $k=30$ and $T=0.3$ essentially correspond to a threshold value of 0.3. As can also be seen from Suppl. Fig. 8, there is (as expected) a strong linear relationship with slope 1 between r_w and $|r_{ws}|$ (and hence between r_w and $|\vartheta|$ for pairs with $r_w > 0.3$). Because we subsequently use a threshold value $|\vartheta| \geq 0.6$, the choice of T has very little impact on the outcome unless one approaches 0.6. We have added this explanation to the legend of Suppl. Fig. 8 (lines 1173-1185).

P. The point here is that when comparing EAP/DAP “by eye”, we intuitively look more at, i.e. give more weight to, the resemblance between the “peaks” of the association patterns. Indeed, we intuitively assume that the ranks of $-\log(p)$ values will be the most stable and hence informative for the highest values. This is the reason why we introduced a weighted correlation. The question of course is what weight to use? When p is 1, w_i corresponds to the ratio between the $-\log(p)$ value of the considered SNP and the top SNP. Thus, a SNP that has a $-\log(p)$ value that is half that of the top SNP receives a weight of 0.5. We just want to show that one can easily modulate this weight by raising the ratio to the power p to either increase or decrease the weight of the top SNPs. At this point we don't know in detail what the effect is of changing p (beyond the scope of this study). However, what we know is that with the parameters chosen, we have a false positive rate of ~ 0.05 and a false negative rate of ~ 0.23 .

Taken together, we consider that our method is robustly defined, and that we know that it produces valid results.

The notion (line 99) that snps are enriched in multi-genic modules again could result from confounding by LD, as surely multi-genic cRM correspond to longer regions of LD. I don't think that the approach detailed in section 7 of supplementary material adequately addresses this.

This is a valid comment. Its prediction is that the level of LD would be higher in GWAS identified IBD risk loci than in the rest of the genome. We have tested this by downloading the 1000-genomes-genetic-maps from <https://github.com/joepickrell/1000-genomes-genetic-maps>.

From this, we computed (i) the average LD-based recombination rate across the entire genome (1.23 cM/Mb), (ii) the average cM/Mb for the 63 IBD risk loci with overlapping eQTL (cfr Table 1) (1.38 cM/Mb), and (iii) the average cM/Mb for 1,000 sets of 63 cRM-centered loci (matched for size and chromosomal location with the 63 IBD risk loci) (1.43 cM/Mb). The 63 IBD risk loci were shown (i) to have a lower level of LD when compared to the entire genome, and (ii) to have a level of LD that does not differ significantly from the cRM centred portion of the genome ($p = 0.46$). We can therefore confidently state that the observed enrichment in multigenic cRM is not due to a higher level of LD in IBD risk loci. We have added a section in the main text describing these analyses (lines 277-291), as well as a Suppl. Figure (Suppl. Fig. 10).

Data availability: There is a web portal but where are they submitting the raw genotypes and transcriptomics data so the whole scientific community can benefit from this impressive resource?

As mentioned above, all data will be made available to the community without restrictions upon acceptance of the manuscript. This is now explicitly mentioned in a "Data availability" section (line 489-491).

MINOR CONSIDERATIONS

eQTLs are from 323 samples across 9 cell types. Average was 56, with a range 19-86, across which the immune system is known to vary. How were age effects considered? Also ascertainment bias - the biopsies were a screening process but I assume the cohort must have had some pre-existing + to undergo such invasive surgery. Information about this needs to be provided.

Sampling. As mentioned in Suppl. Methods (lines 89-96), the participants "*visited the clinic as part of a national screening campaign for colon cancer (free of charge). Enrolled individuals were not suffering any autoimmune or inflammatory disease and were not taking corticosteroids or non-steroid anti-inflammatory drugs (with the exception of low doses of aspirin to prevent thrombosis).*" Approximately 4 of 5 visiting individuals were excluded based on these criteria. We have now added a sentence in the main text to explain that the participants visited the clinic as part of a screening campaign for colon cancer (lines 156-157).

Why use SIFT and Pphen2 what about an ensemble method such as CADD?

We have rerun the burden test using CADD scores instead of SIFT and PPhen2. Following Luo et al. (Nat Genet 49, 186-192, 2017) we used a threshold value of 21 ('CADD21"). By doing so, 78/991 variants predicted by SIFT/PPhen2 to be damaging were ignored, while 573/1,576 variants predicted by SIFT/PPhen2 to be benign were included in the analysis. As shown in Fig. A, there was a strong correlation between SIFT/PPhen2 status and CADD score. Fig. B shows the results of the burden tests. As in the previous analyses, none of the newly tested genes exceeded the Bonferroni corrected $-\log(p)$ threshold. Across genes, the shift towards lower p-values had approximately the same significance, i.e. 7×10^{-4} . However, the order of the genes was slightly changed. We have added a sentence in the main text stating that the results were nearly identical when using CADD scores (lines 335-337), and have added the corresponding results in Suppl. Table 8.

Line 80 - how are the genes 'closely linked' - do the authors mean within the same linkage disequilibrium block? If so how do they define an LD block?

We just mean that they are sufficiently closely located that it is reasonable to hypothesize that they may share the same cis-acting regulatory elements. There is no notion involved here about recombination. We have rephrased using "neighbouring genes" (line 169).

Line 88 - multi-tissular please change to multi-tissue.

Has been corrected.

Line 98 - Can you compute a statistic for whether this sharing between cell types is significant. Supp Fig 2B would be much easier to interpret as a heatmap or as a

hierarchical cluster plot (which would also be a nicer demonstration of reconstituting ontogenic relations).

We have completely reworked this part. As suggested by the reviewer, we have devised a statistical test to evaluate the enrichment of cRM sharing between pairs of cell types (Suppl. Methods, lines 277-297). We describe the corresponding results in the main section of the manuscript (lines 207-215) as well as in (now) main Fig. 1E.

Line 112 - The website is a useful tool however I felt it could do with a more complete help section. The Karyotype browser is difficult to interpret as there are no labels that have biological meaning outside the arbitrary CRM identifiers - perhaps convert to karyotype bands (as human geneticists will have more chance to interpret these), I discourage the use of gene names as this can mislead as to which is the causal gene.

We have added a help section to the CEDAR website that is attached at the end of this rebuttal.

Line 123 - I think that the statement that the burden of rare disruptive variants is expected to differ between cases and controls needs a citation to back it up. For example, in T2D Fuchsberger, C. et al. (2016) The genetic architecture of type 2 diabetes. Nature.

This reference has been added.

Line 130 - What is the allelic spectrum of the variants found? How do these compare with GNOMAD.

Of the 4,175 variants detected in this study, 1,781 were also reported in GNOMAD (Lek et al., 2016). As shown in the figure, the frequency of the alternate allele in our study versus in the GNOMAD 55,860 non-Finnish Europeans was nearly identical.

We have added a sentence in the main text to highlight this (lines 308-310).

Line 152 - What is superscript f.i.3 ?

“f.i.” has been eliminated.

Line 203 - I think the evidence for these conclusions is weak also the authors should say which statistical test was performed.

This is the p value obtained with CAST. It is now mentioned in the text (line 383).

How to use the CEDAR website?

1. Select your disease of interest by activating the corresponding link in the home page (<http://cedar-web.giga.ulg.ac.be/index.html>).

Nearly all common complex diseases are heritable: we are not all equally susceptible to the environmental and behavioural risk factors that trigger them, and these inter-individual differences are in part genetic. GWAS conducted in the last decade show that this heritability is driven by hundreds to thousands of genetic variants scattered across the genome. A minority of those are coding variants that act by changing the amino-acid sequence of the encoded protein. The majority are regulatory variants, which act by perturbing gene switches thereby altering the expression profile of one or more target genes. Such variant-dependent perturbations of gene expression are also known as expression Quantitative Trait Loci (eQTL).

SELECT A DISEASE:
Crohn's Disease (CD) >
Ulcerative Colitis (UC) >

To aid in the identification of the causative genes that are perturbed by disease-causing regulatory variants, we have generated a transcriptome data set of nine disease relevant cell types (CD4, CD8, CVD19, CD14, CD15, platelets, ileonic, colonic and rectal biopsies) in 300 healthy European individuals. Using this data, we have identified 24,000 cis-eQTL reflecting the effects of 9,700 distinct regulatory modules (constellations of regulatory variants). We have developed a metric, to identify cis-eQTL that match disease association patterns (DAP) and could therefore drive the disease. The CEDAR website provides a genome-wide overview of the corresponding results for important common complex diseases.

Funding CEDAR: CEDAR was funded by grants to Michel Georges from WELBIO (CAUSIBD), BELSPO (BeMGI), and Horizon 2020 (SYSCID).

Citing CEDAR: Momozawa et al. Risk loci for inflammatory bowel disease are enriched in multigenic regulatory modules that act across cell types and encompass causative genes. Submitted for publication.

Data visualisation by PolyOmica

2. This leads you to the following "Genome View".

Select a disease (CD) / Select a risk locus

The Genome View ↔ Switch to the Table View

	Region	Chr	From	To
1	N_6_0	6	0	882559
2	HD58	5	439518	778710
3	J_1_0	1	990000	1490000
4	HD165	19	1041164	1225200
5	J_11_0	11	1620000	2120000
6	HD1	1	2357159	2835238
7	J_7_1	7	2530000	3030000
8	N_6_2	6	2920406	3920406
9	N_4_2	4	2944503	3944503
10	HD102	9	4894106	5323847
11	HD107	10	5980257	6219615
12	N_12_5	12	5991125	6991125
13	HD2	1	7688668	8254206
14	HD166	19	10346439	10677557
15	J_5_9	5	10440000	10940000
16	HD149	16	11263407	11527821
17	J_12_11	12	12400000	12900000
18	HD162	18	12690029	12972583

Select a risk locus of interest by clicking on the corresponding dot.

3. This leads you to the following “Risk Locus View”.

Select a disease (CD) / Select a risk locus (HD35) / Select a probe

The Genome View ↔ Switch to the Table View

You can scroll/zoom through the locus by changing the size of the blue rectangle at the top. The triangles show the position of the gene-specific probes from the Illumina HT12 array. Filled dots mark the existence of cell-type specific eQTL. Black dots correspond to eQTL that show no correlation with DAP ($|\vartheta| < 0.6$). Red and green dots correspond to eQTL that are correlated with DAP ($\vartheta \geq 0.6$ and $\vartheta \leq 0.6$, respectively).

Select an eQTL by clicking on an eQTL dot.

4. This leads you to the following “eQTL or Probe view”.

Select a disease (CD) / Select a risk locus (HD35) / Select a probe

Region: HD35
 Tissue: CD4
 Gene name: IL18R1
 Probes: 1570367
 Cor: 0.93 (P: 0.010)
 Signed cor: -0.93 (P: 0.010)

It shows two graphs. The left one provides DAP (black dots) and EAP (red dots) including gene positions. The right one shows the relationship between the $-\log(p)$ values of the DAP and EAP (adjusted for the signs of the effects – see Momozawa et al., 2018) from which ϑ is computed.

At the top, the “Probe view” provides information about (i) the risk locus, (ii) the cell type, (iii) the gene, (iv) the probe, (v) the values of r_w (“corr”) and ϑ (“signed corr”) and corresponding empirical p-values (as defined in Momozawa et al., 2018).

5. An alternative way of using the CEDAR website is via the “Table View” accessible from the “Genome View” and “Risk Locus View”.

cedar Correlated Expression & Disease Association Research

Select a disease (CD) / Select a risk locus and a gene

The Table View ↔ Switch to the Genome View

Maximize

Filter the table:

Select a Tissue:

Select a Region:

Select a Gene:

$-\log_{10}(\text{eQTL_Pval})$ filter:

$-\log_{10}(\text{P_Sign_Corr})$ filter:

$-\log_{10}(\text{P_Corr})$ filter:

Corr filter:

Corr_Sign filter:

	Region	Tissue	Gene	SNP	BP	eQTL_Pval	eQTL_Qval	Corr	
1	↗	HD35	CD4	IL18R1	rs11123923	102967844	6.418e-27	0.00027808	0.93
2	↗	HD35	CD8	IL18R1	rs11123923	102967844	4.697e-14	0.00035446	0.91
3	↗	HD35	CD15	IL18R1	rs1041973	102955468	0.000001575	0.0054948	-0.55

It allows you to query the CEDAR database using a variety of filters. You can then access the corresponding “eQTL/Probe view” by following the corresponding ↗ link. From there you can access the corresponding “Risk locus view”.

Reviewer #1 (Remarks to the Author):

The authors have done a great job in addressing all the reviewers' concerns and the readability of the manuscript has improved significantly. The presented data is a great add-on to the fine mapping manuscript of the IIBDGC. I only have two minor issues and one one last question:

minor issue 1) When the authors mention the surface markers for sorting the cells they should also write out what cell type are selected. Not every author is familiar with the CD-nomenclature.

minor issue 2) Please define early on the term cis-eQTL and say what trans-eQTLs are.

Last question: I am surprised that the authors have not detected many signals in the HLA/MHC region on chromosome 6p21. While this is one of the most important loci for autoimmune diseases, it is also one of the most complex. What is the coverage of the classical HLA alleles on the HT12 array? At least the biopsies and epithelial cells should express these HLA molecules and transcripts.

Reviewer #3 (Remarks to the Author):

I have only two major issues remaining:

- the lack of comparison with previous analyses of these same data, particularly Huang et al (2017) which used the same GWAS and eQTL data to find modest overlap, vs this paper which finds overlap at 63 loci. I appreciate details are given in the rebuttal, but those have not made it through to the manuscript in required detail. This is important for readers of the IBD literature who will see two analyses of the same data by the same authors reaching very different conclusions and will need to understand why.

- the lack of data availability. It is not enough to state the data will be made available. Such statements are frequently ignored after publication and editors do not enforce them. Without casting any aspersions on the open-data willingness of these authors, they need to be uploaded to a repository, and the URL given, to ensure confidence in the reproducibility. It is also particularly important as one of the primary outcomes of this work is, imo, this body of multiple cell type eQTL data.

I note also that this statement - that the data will be available without restriction - is in direct conflict with the data availability statement for the Huang paper which says the GWAS data are only available under license but doesn't mention the eQTL data.

Please pay more than lip service to data availability.

Reviewers' comments:

Reviewer #1 (Remarks to the Author):

The authors have done a great job in addressing all the reviewers' concerns and the readability of the manuscript has improved significantly. The presented data is a great add-on to the fine mapping manuscript of the IIBDGC. I only have two minor issues and one last question:

minor issue 1) When the authors mention the surface markers for sorting the cells they should also write out what cell type are selected. Not every author is familiar with the CD-nomenclature.

We have added the names of the corresponding cell types, both in the main text (lines 155-156), as well as in Suppl. Materials (lines 123-124).

minor issue 2) Please define early on the term cis-eQTL and say what trans-eQTLs are.

We have now added a definition of cis- versus trans-eQTL (lines 110-116).

Last question: I am surprised that the authors have not detected many signals in the HLA/MHC region on chromosome 6p21. While this is one of the most important loci for autoimmune diseases, it is also one of the most complex. What is the coverage of the classical HLA alleles on the HT12 array? At least the biopsies and epithelial cells should express these HLA molecules and transcripts.

The HT12 array has 38 probes targeting 28 HLA genes on 6p21. We found 66 significant cis-eQTL (FDR < 0.05) affecting 13 HLA genes in all nine analyzed cell types tissues (see Suppl. Table 1). Only one of these cis-eQTL (HLA-DQA2 in ileum) matched the DAP for UC (see Table 1) with $|\rho| \geq 0.6$. We surmise that, in the case of HLA, many of the causative variants are coding and would therefore not generate matching DAP-EAP.

Reviewer #3 (Remarks to the Author):

I have only two major issues remaining:

- the lack of comparison with previous analyses of these same data, particularly Huang et al (2017) which used the same GWAS and eQTL data to find modest overlap, vs this paper which finds overlap at 63 loci. I appreciate details are given in the rebuttal, but those have not made it through to the manuscript in required detail. This is important for readers of the IBD literature who will see two analyses of the same data by the same authors reaching very different conclusions and will need to understand why.

As mentioned in our previous response to the reviewers, we tried to accurately address the causes of the difference between Huang et al. and this study (in terms of number of matching DAP-EAP and significance of the enrichment) in the main

text (lines 255-266), in Suppl. Methods (lines 316-379) and in Suppl. Table 6. We have now added a sentence to the legend of Table 1 to clearly state what factors underlie the differences in numbers between the two studies (lines 597-600).

- the lack of data availability. It is not enough to state the data will be made available. Such statements are frequently ignored after publication and editors do not enforce them. Without casting any aspersions on the open-data willingness of these authors, they need to be uploaded to a repository, and the URL given, to ensure confidence in the reproducibility. It is also particularly important as one of the primary outcomes of this work is, imo, this body of multiple cell type eQTL data.

I note also that this statement - that the data will be available without restriction - is in direct conflict with the data availability statement for the Huang paper which says the GWAS data are only available under license but doesn't mention the eQTL data.

Please pay more than lip service to data availability.

As stated in the “data availability” section (lines 485-487), the entire CEDAR eQTL dataset, including (i) the SNP genotypes (real and imputed), (ii) the expression data of all genes in all tissues (both raw and pre-processed) will be made freely available without any restrictions. The data will be accessible from an ftp site linked to the CEDAR website, as well as from then European Genome-Phenome Archive (EGA). We are in the process of organizing this and transferring the data.